**METHOD**

# Melon: metagenomic long-read-based taxonomic identification and quantification using marker genes

Xi Chen[1], Xiaole Yin[1], Xianghui Shi[1], Weifu Yan[1], Yu Yang[1], Lei Liu[1] and Tong Zhang[1*]

*Correspondence:
zhangt@hku.hk

[1] Environmental Microbiome Engineering and Biotechnology Lab, Department of Civil Engineering, The University of Hong Kong, Pokfulam Road, Hong Kong, China

## Abstract

Long-read sequencing holds great potential for characterizing complex microbial communities, yet taxonomic profiling tools designed specifically for long reads remain lacking. We introduce Melon, a novel marker-based taxonomic profiler that capitalizes on the unique attributes of long reads. Melon employs a two-stage classification scheme to reduce computational time and is equipped with an expectation-maximization-based post-correction module to handle ambiguous reads. Melon achieves superior performance compared to existing tools in both mock and simulated samples. Using wastewater metagenomic samples, we demonstrate the applicability of Melon by showing it provides reliable estimates of overall genome copies, and species-level taxonomic profiles.

**Keywords:** Long-read sequencing, Taxonomic profiling, Metagenomics

## Background

Characterization of microbial communities colonizing a particular ambient is central for studies that aim to comprehend the ecological, pathological or functional roles of species under different environmental conditions [1–3]. While conventional methods for identification and quantification of microbial species in communities (referred to as "taxonomic profiling") typically rely on cultivation of isolated organisms or amplification of the 16S ribosomal RNA (rRNA) genes [4], modern methods predominantly opt for high-throughput whole-genome shotgun sequencing (WGS) due to its ability to render an unbiased genomic snapshot of all organisms in a given sample [5]. A key component of WGS-based methods is to taxonomically classify randomly generated genomic fragments (also called sequences of base pairs or simply reads) with a reference database. Depending on the type of the database employed, these classification methods can be broadly categorized into two groups: (1) DNA-to-DNA/protein and (2) DNA-to-marker taxonomic profilers [6, 7].

DNA-to-DNA/protein profilers attempt to assign taxonomic labels to the entire collection of sequencing reads through direct comparisons with comprehensive databases that consist of nucleotide or protein sequences (e.g., RefSeq genomes or RefSeq proteins) [8]. Examples within this group include Kraken2 (and its related tools such as Bracken and Kraken2Uniq) [9–11], Centrifuge [12], Kaiju [13], MEGAN [14], as well as commonly used alignment tools, though not developed specifically for taxonomic assignment: BLAST [15], MMseqs2 [16], DIAMOND [17], and minimap2 [18]. These profilers, despite largely different strategies utilized for database construction, read mapping, and result aggregation, by default all report the relative abundance of a given taxon as the proportion of reads binned to it in relation to the total number of reads sequenced, i.e., sequence (relative) abundance [6, 19]. In contrast, DNA-to-marker profilers, such as MetaPhlAn4 [20] and mOTUs3 [21], only assign taxonomic labels to a small fraction of sequencing reads that map to databases comprised of specific gene families (i.e., phylogenetic marker genes). Leveraging the universally distributed and single-copy nature of these marker genes, DNA-to-marker profilers typically output the relative abundance of a given taxon as the genome copy of that taxon divided by the total genome copy detected, i.e., taxonomic (relative) abundance [6, 19, 22, 23].

In comparison with sequence abundance, taxonomic abundance represents the fraction of cells (assuming one genome copy per cell) rather than reads for each detected taxon [6, 23]. This feature allows it to deliver more biological insights and makes it more practically useful for studies that require knowing the genomic coverage of individual taxon, e.g., absolute quantification with external cellular spike-ins, or measuring the diversity using abundance-aware indexes, e.g., Shannon's diversity index [6, 24]. Furthermore, although theoretically sequence abundance can be translated into taxonomic abundance via proper genome size correction, the reality often proves challenging as a significant number of species to date still lack complete genome representatives, thereby inhibiting the construction of precise genome size databases [6, 8, 24]. This again underscores the importance of DNA-to-marker methods, which directly yield profiles of taxonomic abundance.

In fact, while taxonomic abundance confers multiple advantages, DNA-to-marker methods have received considerably less attention over the past years and been significantly outnumbered by DNA-to-DNA/protein methods [6]. This is particularly evident when third-generation sequencing (also known as "long-read sequencing") is being considered [22]. Long-read sequencing, with its capacity to generate much longer yet comparably accurate reads to short-read sequencing, is expected to substantially enhance taxonomic resolution when integrated with taxonomic profiling. [25, 26]. Thus far, a variety of taxonomic profilers tailored specifically for long reads, such as MetaMaps [27], MEGAN-LR [28], and BugSeq [29], have already been developed. However, all these tools are inherently DNA-to-DNA/protein [27–29].

Currently, no long-read targeting DNA-to-marker taxonomic profiler is available [22]. Although mOTUs3 recently introduced a preprocessing step that enables it to handle long reads by fragmenting them into short reads, it cannot be regarded as a native long-read method due to the loss of long-range information [21, 22]. To bridge this gap, we developed Melon, a new DNA-to-marker taxonomic profiler that capitalizes on the unique attributes of long-read sequences. Taking advantage of a two-step classification

scheme, Melon is able to (1) estimate total prokaryotic genome copies and (2) provide species-level taxonomic abundance profiles in a fast and precise manner. In particular, this is achieved by first extracting marker-containing reads and subsequently mapping them to a size-reduced RefSeq genomes database via highly accurate alignment-based approaches. Using both in vitro mock and in silico simulated samples, we showed that Melon outperforms existing tools: mOTUs3, Kraken2, and Bracken (with genome size correction applied to Kraken2 and Bracken), in both species identification and quantification. We further demonstrated the applicability of Melon using two real-world wastewater samples by showing its capability of providing species-resolved, biologically meaningful normalizing constants for antibiotic resistance genes (ARGs)—a leading public health threat—in complex settings, thereby facilitating downstream ARG profiling, comparison, and host-tracking.

## Results

### Melon overview

Given a sequenced metagenomic sample, Melon first extracts reads that cover at least one marker gene using a protein database, and then profiles the taxonomy of these marker-containing reads using a separate, nucleotide database (Fig. 1a). The use of two different databases is motivated by their distinct strengths: the protein database is particularly well-suited for estimating the total number of genome copies because of its high conservation, whereas the nucleotide database has the potential to provide a greater taxonomic resolution for individual reads during profiling [6].

We initiated the construction of the aforementioned protein database by re-annotating NCBI non-redundant protein sequences (nr) and metagenomic proteins (env_nr) using 91 profile hidden Markov models (PHMMs) (Fig. 1b, Additional file 1: Data S1, Methods). These PHMMs were collected from KOfam (a customized PHMM database of KEGG orthologs with pre-computed adaptive score thresholds) [30], and corresponded to the 55 bacterial and 67 archaeal ribosomal protein genes (RPGs) documented in "ribosomal protein gene clusters in prokaryotes" (https://www.genome.jp/kegg/annotation/br01610.html, accessed July 31, 2023). We exclusively selected RPGs as they are characterized by low mutation rates owing to strong evolutionary constraints [31], and are more likely to be essential due to their fundamental role involved in protein synthesis [32, 33]. A subset of RPGs were screened as marker genes (eight each for both bacteria and archaea, see the "Quality assessment of PHMMs and RPGs" section for more details) by assessing the quality of each according to four distinct factors: (1) $F_{0.5}$-scores (weighted harmonic mean of precision and recall) of their associated PHMMs, (2) their prevalence among species, (3) discrepancies between their average numbers of copies and the ideal count of one, and (4) their mean relative genomic distances to other candidate RPGs (Fig. 1c). Protein sequences of nr and env_nr that aligned with the PHMMs of these marker genes were retained for further analysis. After deduplication through clustering, the final protein database comprised 468,432 unique sequences.

The nucleotide database intended for assigning taxonomic labels was built by extracting 10,000 bp genomic regions encompassing marker genes and their adjacent flanking regions from 310,881 assemblies (corresponding to 44,057 bacterial and 1,016 archaeal species) collected from NCBI RefSeq as of July 31, 2023 (Fig. 1d, Additional file 1: Data

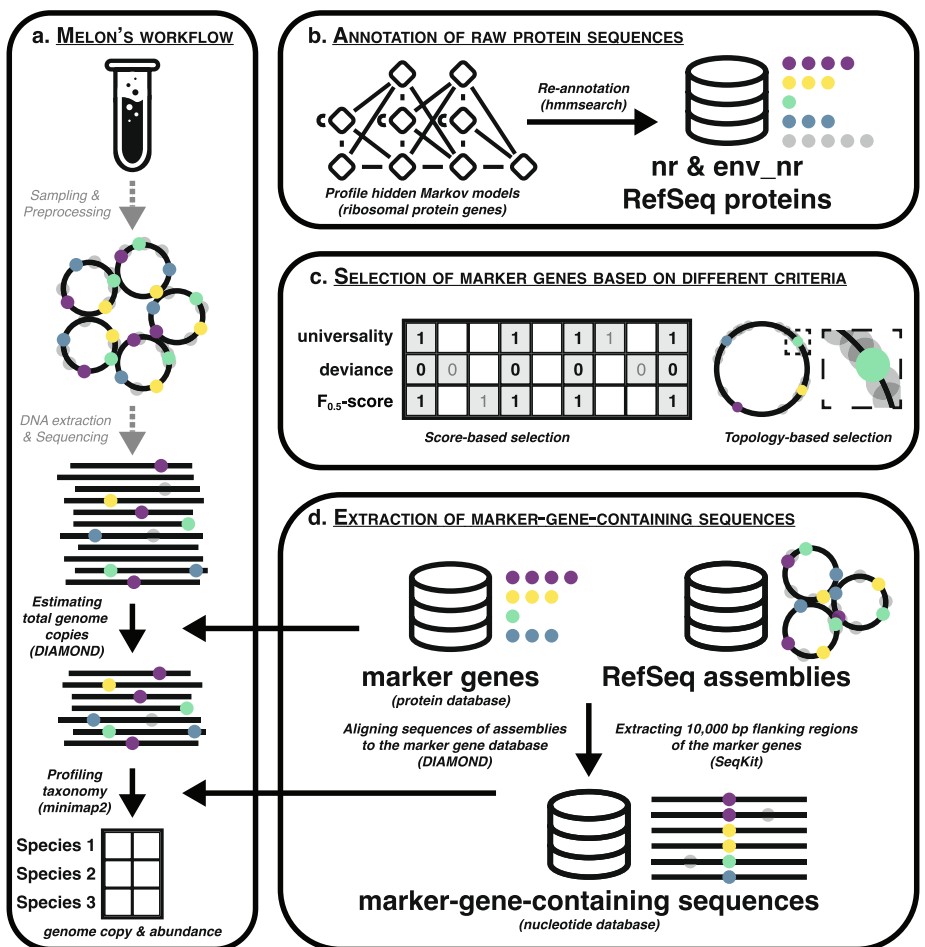

**Fig. 1** Overview of Melon. **a** Melon's workflow. Given a sequenced metagenomic sample, Melon first extracts reads that cover at least one marker gene using a protein database (DIAMOND), and then profiles the taxonomy of these marker-containing reads using a nucleotide database (minimap2). The main output of Melon is a tab-delimited table listing the estimates of species' genome copies and relative abundances. Gray dashed arrows and text indicate necessary sample preprocessing steps to obtain metagenomic long reads. **b** The construction of the protein database is initiated by re-annotating NCBI protein sequences using hmmsearch and a set of RPG-related PHHMs. **c** A subset of RPGs are selected as marker genes based on their universality, deviance, $F_{0.5}$-scores, and mean relative genomic distances. **d** The nucleotide database is built by extracting 10,000 bp genomic regions, which encompass marker genes and their adjacent flanking regions, from 310,881 RefSeq assemblies using DIAMOND and SeqKit. These assemblies represent 44,057 bacterial and 1016 archaeal species. **a–d** Solid colored dots stand for marker genes, while semi-transparent gray dots represent other genes. Circular elements denote genomes, while linear elements signify reads

S2) [34]. To reduce redundancy, we clustered the extracted sequences for each species and marker gene combination, achieving a data compression ratio of over 100 (from approximately 1,304.2 billion bp and 32.6 million sequences to 8.9 billion bp and 0.9 million sequences). This substantial decrease in the size of the reference database rendered computationally intensive alignment-based taxonomic labeling feasible on a standard laptop (Additional file 2: Table S1). In addition to the RefSeq database, we also constructed a nucleotide database that followed the nomenclature and taxonomic classification of the Genome Taxonomy Database (GTDB) (release 08-RS214) using 402,690 assemblies (with 19 assemblies being deprecated by NCBI and thus not included), to facilitate different usage preferences [35].

**Quality assessment of PHMMs and RPGs**

The quality of the 91 PHMMs and their corresponding RPGs was evaluated using prokaryotic RefSeq proteins and "complete genome" RefSeq assemblies ($n = 35,224$), respectively [34]. To alleviate the overrepresentation issue of certain lineages (e.g., *Escherichia coli* had 2834 assemblies), when multiple assemblies were available for a given species, only the one with the highest quality score (defined as completeness minus five times contamination, predicted by CheckM2 [36]) was retained [37]. This resulted in a total of 9195 assemblies, among which 234 exhibited highly reduced genomes (less than 1 Mb). We explicitly considered assemblies of tiny genomes as a secondary validation set, as evidence suggests that certain RPGs might be nonessential for host-associated prokaryotes (e.g., symbionts and parasites with drastically reduced genomes) and susceptible to loss in the course of genome compaction [32]. Conversely, RPGs persisting even in condensed genomes are more likely to be indispensable for survival, making them ideal candidates as marker genes [33].

Out of the 91 PHMMs, 45 bacterial and 23 archaeal PHMMs had $F_{0.5}$-scores greater than 0.99 (Fig. 2a). The mean $F_{0.5}$-scores were 0.977 and 0.809 for bacterial and archaeal PHMMs, respectively (Additional file 1: Data S1). Note that we scaled the pre-computed threshold scores to a factor of 0.75 consistently while annotating sequences, since we observed that the default threshold scores were relatively strict, especially for archaeal sequences (Additional file 2: Table S2) [30]. Regarding the assessment of RPGs, we computed for each RPG its universality (defined as the proportion of assemblies containing it) and deviance (defined as the absolute difference between its mean copy number and the ideal count of one) using (1) the entire collection of assemblies ($n = 9195$) and (2) the tiny genome subset ($n = 234$). Overall, 21 bacterial and 38 archaeal RPGs met the criteria of having universality greater than 0.99 and deviance less than 0.01 in both sets of assemblies, with 21 bacterial and 17 archaeal RPGs among them also satisfying the requirement of having a PHMM's $F_{0.5}$-score exceeding 0.99 (Fig. 2a). Some losses of RPGs appeared to be strongly size-specific. For instance, despite an overall universality of 0.990, the bacterial RPG l29 was not identified in multiple size-reduced assemblies, causing a much lower universality of 0.822 in the tiny genome subset (Additional file 1: Data S3).

For the 21 bacterial and 17 archaeal RPGs, we calculated their pairwise shortest genomic distances within each of the available circular genome assemblies ($n = 8589$). These distances were then normalized according to the sizes of their respective genomes and subsequently averaged to create a mean relative genomic distance matrix. Based on the distance matrix, we identified RPGs that consistently collocated (Fig. 2b). These RPGs were grouped into eight clusters for both bacteria and archaea, as closely situated RPGs do not provide additional insights into the variation in sequencing coverage caused by bidirectional replication (from a fixed origin to a fixed terminus) of prokaryotic genomes [38]. Furthermore, for fast-growing species, genome copies might be significantly overestimated or underestimated if marker genes are centered around replication origins or termini (Fig. 2c). We selected a single representative RPG from each cluster to serve as a marker gene. This led to a total of eight marker genes for both bacteria and archaea.

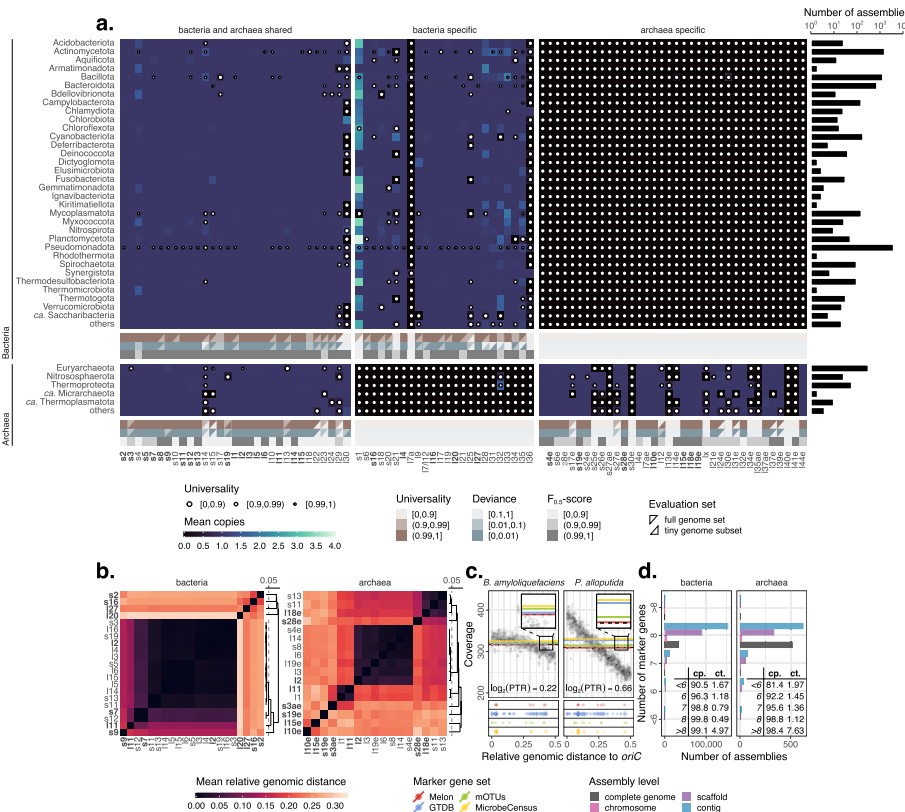

**Fig. 2** Selection of marker genes. **a** Mean copies, universality and overall performance of RPGs. Sizes of white dots represent the values of universality, with universality of one being omitted. Overall universality, deviance, and $F_{0.5}$-scores of RPGs are shown below the heatmaps of mean copies. Shapes of triangles indicate evaluation sets, with top-right triangles representing the full set of assemblies, and bottom-right triangles the tiny genome subset. RPGs satisfying all three criteria (darkest colors in all three rows) for either bacteria or archaea are marked in bold. The bar chart on the right side shows the number of assemblies (species) present in each phylum. Phyla with only a single assembly are grouped under "others." **b** Clustering of valid RPGs based on their mean relative genomic distances. Selected RPG representatives of bacteria and archaea are marked in bold. **c** Effects of marker genes' locations on genome copy estimations. Black dots represent sequencing coverage at different genomic locations, smoothed using 5000 bp non-overlapping windows. Colored dots display the genomic locations of marker genes in four sets: Melon (8 genes), GTDB (120 genes), mOTUs (10 genes), and MicrobeCensus (30 genes). Dashed black lines denote expected genome copies, whereas solid colored lines stand for estimated genome copies produced by averaging the coverage of marker genes in different sets, smoothed again using 5000 bp windows. Melon's genome copy estimates are the closest to the expected genome copies of both *Bacillus amyloliquefaciens* (*B. amyloliquefaciens*, Gram-positive) and *Pseudomonas alloputida* (*P. alloputida*, Gram-negative), owning to its relatively balanced marker gene distributions. Log2-transformed peak-to-trough ratios (PTRs) are displayed in text. *oriC* means the origin of replication. **d** Numbers of marker genes detected in RefSeq assemblies. Colors indicate assembly levels. Completeness (cp.), contamination (ct.), and their relations to the number of marker genes are shown in tables

We then extracted sequences of at most 10,000 bp that covered the selected marker genes from RefSeq assemblies. After deduplication, we computed their pairwise average nucleotide identities (ANIs) using skani [39]. The median ANI ranged from 99.0% (bacterial s16) to 99.8% (bacterial l2) between strains and from 93.8% (archaeal l2) to 96.9% (archaeal l10e) between species. Given that RPGs are in general more conserved compared to other nonessential genes, it was expected that some between-species ANIs exceeded the conventional ANI cutoff for species identification, i.e., 95%. However, for

most marker genes, there was still a gap between the within-species and within-genus ANIs. This large difference in species/genus-level ANIs provided evidence that these marker-gene-containing sequences were sufficient for delivering species-level taxonomic resolution (Additional file 2: Note S1).

Of the 310,881 RefSeq assemblies, 284,039 (91.4%) possessed precisely eight unique marker genes. Assemblies missing one or more marker genes tended to have lower completeness, while those with over eight marker genes were likely contaminated (Fig. 2d). This trend of missing or excessive marker genes was particularly pronounced in fragmented assemblies: 99.0% of bacterial and 97.8% of archaeal complete genome assemblies contained exactly eight unique marker genes, whereas these percentages dropped to 90.1% and 80.0% for contig level assemblies.

## Performance evaluation using mock and simulated samples with different levels of complexity

### *Mock experiment*

To assess the performance of Melon in relation to other tools, we collected six synthetic mock samples. These samples were all generated by Oxford Nanopore Technologies (ONT) devices but with different sequencing chemistries and basecalling models, thereby exhibiting a broad range of quality scores and read lengths (Additional file 2: Table S3). Three samples contained eight bacteria in even sequence abundances and additionally two yeasts (S1–3, ZymoBIOMICS Microbial Community Standard, D6300), whereas the remaining three had staggered sequence abundances of fourteen bacteria, one archaeon, and two yeasts (G1–3, ZymoBIOMICS Gut Microbiome Standard, D6331). Expected values of relative abundances and genome copies were obtained for individual species by mapping reads to their reference genomes. We did not use the theoretical relative abundances provided by ZymoBIOMICS as they might differ from the expected ones due to operational variation during library preparation (Additional file 2: Fig. S1) [40]. Reads that mapped to the reference genomes of yeasts or remained unmapped were discarded to avoid overestimation in genome copies caused by eukaryotes [41], and potential contamination by e.g., barcode "crosstalk" [42].

The set of tools used for comparison included Melon (v0.1.0), Kraken (v2.1.3), mOTUs (v3.1.0), and Kraken's companion tool Bracken (v2.8). For Kraken and Bracken, genome size correction is needed to convert sequence abundance into taxonomic abundance [6]. We thus built a hierarchical genome size database utilizing all available RefSeq complete genome assemblies, and employed the genome size of the lowest taxonomic level for correction whenever possible. For Bracken, since taxonomic assignments of reads are not given, we assumed all reads had a length equal to the sample's average length.

We evaluated the performance based on three criteria: (1) accuracy of the estimated total genome copies, (2) precision and recall for species-level taxonomic assignments, and (3) distances (or dissimilarities) between the estimated and expected relative abundances (Methods). Among the tools listed above, Melon achieved the best performance in all these aspects, except for recall (Fig. 3a). The low recall is a common limitation of marker-based methods, as only fractions of genomes and reads, specifically those associated with marker genes, are involved in taxonomic labeling. Consequently, an insufficient sequencing depth may result in missed species. However, with increased

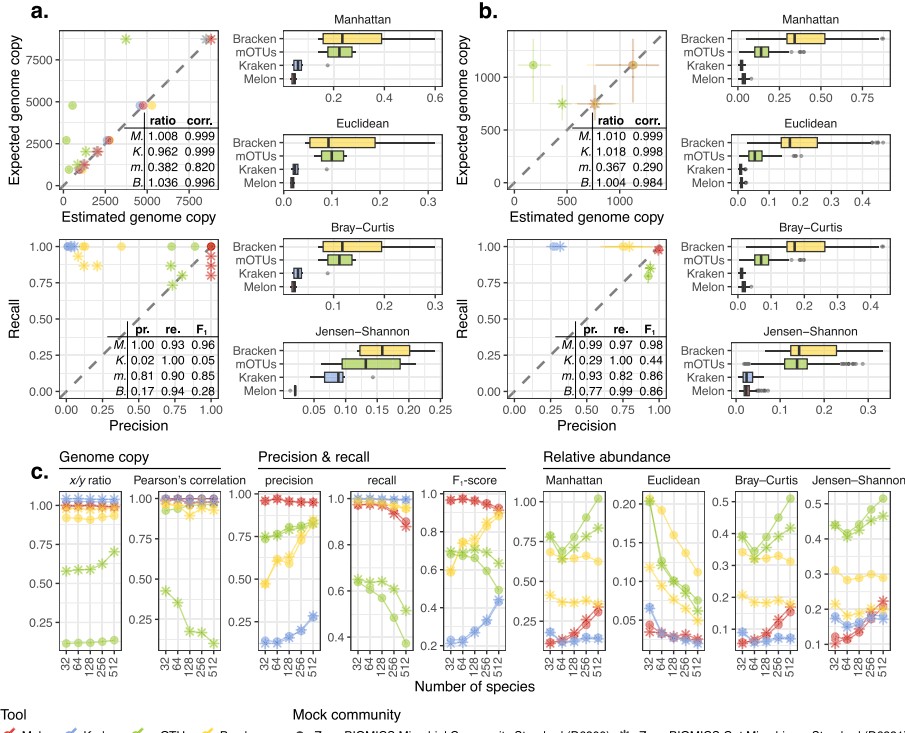

**Fig. 3** Performance of Melon on mock and simulated samples. **a** Mock experiment (mock samples S1–3 and G1–3). **b** CAMI experiment (64 mouse gut profiles). **c** Complexity experiment (5 × 10 artificial profiles). **a**–**c** Colors indicate tools used for comparison, including Melon (v0.1.0, NCBI database, ver. 2023-07-31), Kraken (v2.1.3, Standard database, ver. 2023-06-05), Bracken (v2.8, same database as Kraken) and mOTUs (v3.1.0, default database, with modifications). Shapes of points denote different mock communities, D6300 and D6331. Tables provide summarized metrics, including ratio (*x/y* ratio), corr. (Pearson's correlation), pr. (precision), re. (recall), and $F_1$ ($F_1$-score). Abbreviations used for tools' names are M. (Melon), K. (Kraken), m. (mOTUs), and B. (Bracken). Error bars depict the ranges of possible values. All metrics, except for Pearson's correlation, are aggregated by averaging where necessary

sequencing depths at 3.6 Gb, 6.5 Gb, and 27.9 Gb, we observed that 12, 13, and 14 out of 15 species of the D6331 dataset, respectively, could be detected—even with a minimal expected relative abundance down to $1.4 \times 10^{-5}$ and a genome copy to 0.124. This gradual increase in recall was expected, as Melon's detection limit is independent of relative abundance and is fixed at a genome copy of approximately 0.125, owing to the use of eight marker genes. For mOTUs, the estimated total genome copies differed substantially from the expected ones, especially for the D6300 dataset (which had lower overall quality scores compared to D6331, Additional file 2: Table S3). This low recovery rate of genome copies could largely be attributed to the fact that mOTUs adopts BWA—which is not designed primarily for error-prone sequences—as its backend aligner [43]. In contrast, given proper genome size correction, Kraken and Bracken performed reasonably well in estimating total genome copies. However, as reflected by their inferior *x/y* ratios (ratios between the estimated and expected total genome copies, i.e., recovery rate of genome copies) and Pearson's correlations, the estimates of Kraken and Bracken tended to fluctuate around the expected values. This could be explained by the fact that a minority of reads were classified at levels above species or remained unclassified, leading to less accurate genome size correction. Moreover, the multicopy nature of plasmids might

also complicate the correction. Kraken achieved the highest recall but the lowest precision. Bracken slightly improved upon Kraken's precision through Bayesian re-estimation, at the cost of recall. mOTUs, on the other hand, was capable of providing balanced precision and recall, yet its overall performance, including $F_1$-score (unweighted harmonic mean of precision and recall), was still lower in comparison to Melon. In terms of relative abundance, Melon's estimates showed the highest concordance to the expected values, regardless of the metric used. It is worth noting that this occurred even though some species in the mock samples were actively growing and exhibited uneven sequencing coverage along their genomes. Kraken also produced reasonable relative abundance estimates, despite its low precision. Both Bracken and mOTUs failed to provide reliable estimates of relative abundances. Bracken's inconsistency likely stemmed from the assumption that all reads were of equal length, which introduced variance in genome size correction. For mOTUs, the discrepancy was probably due to varied recovery rates of different species, which skewed the abundance estimates.

### CAMI experiment

To see the performance of Melon in more complex settings, we collected 64 mouse gut profiles from the second Critical Assessment of Metagenome Interpretation (CAMI) challenge [19]. For each profile, we simulated six samples (500,000 reads each) using the metagenome mode of NanoSim [44], with models trained specifically on the six mocks to emulate their error distributions and read characteristics (Additional file 2: Table S4). This resulted in a total of 384 samples, with species counts ranging from 9 to 38 and sample sizes from 2.3 Gb to 5.3 Gb. Note that this simulation can be seen as a simplified version of the original, as we kept only species whose genomes were present in the intersection of the three reference databases (Kraken and Bracken share databases) to avoid database-induced biases (Methods) [6].

Evaluation results from the simplified CAMI experiment largely aligned with those of the mock dataset, with Melon again demonstrating the best performance in most aspects (Fig. 3b). Bracken marginally outperformed Melon in $x/y$ ratio, but not in Pearson's correlation. In precision-recall analysis, Melon achieved the highest precision with only a minor drop in recall. The high precision of Melon was likely attributable to the implementation of expectation-maximization (EM) as a post-correction module, which greatly reduced the number of false positive classifications (Additional file 2: Note S2). All other tools showed increased $F_1$-scores, though to varied degrees. This was likely because many species in the simulation dataset were not as common as the ones in the mock dataset, therefore easier to classify. This also explained why Kraken and Bracken had a drastic increase in precision. For relative abundance estimates, Kraken showed slightly better performance compared to Melon, possibly due to its ability to detect more rare species. Nevertheless, the difference in their estimates depended on the metric being examined. For instance, with Jensen–Shannon divergence, the difference was not statistically significant (Wilcoxon test, two-sided, $p = 0.734$).

### Complexity experiment

To further investigate the performance change with increased complexity, we simulated a series of samples using a fixed number of reads (500,000) and a varied number of

species (32–512), with each repeated ten times (Methods). The simulation settings were largely identical to the simplified CAMI experiment, except that species were randomly chosen from a larger pool ($n = 8618$, covered by all three databases) and their abundances followed a log-normal distribution ($\mu = 0$, $\sigma = 1$). It is noteworthy that although we did not explicitly include strains in this simulation experiment, many species were close relatives and could possess highly similar or even exactly identical genomic regions of the marker genes.

In terms of genome copies, Melon consistently yielded stable and accurate estimates, as evidenced by both $x/y$ ratio and Pearson's correlation (Fig. 3c). Kraken and Bracken, on the other hand, exhibited systematic biases in their estimates, and these biases were likely influenced more by the models used for simulation rather than the complexity of the samples. For mOTUs, the trend was less clear: while its Pearson's correlations decreased as complexity increased, $x/y$ ratios showed the opposite trend. This pattern implies that mOTUs has a higher chance of mapping sequences to its reference database when more species are present, yet the variability in recovery rates across species causes more scattered results. Melon's taxonomic classification displayed the highest precision across all settings. Intriguingly, all other tools demonstrated improved precision as complexity increased. This improvement was probably a result of the elevated coverage of reference databases: when evaluating precision based on the presence or absence of species, a misclassified read will not be counted as a false positive if the corresponding species coincidentally exists in the sample. In the most extreme scenario where a sample includes all species of a database, precision will always be perfect, regardless of the tool used. Meanwhile, as the number of species increased, recall declined—a trend we anticipated, as many species became less covered and tended to fall below detection limits. This decreasing trend was especially evident for marker-based methods like Melon and mOTUs, for previously mentioned reasons. When considering precision and recall as a whole, Melon achieved the highest overall $F_1$-scores across all complexity settings. However, Bracken might potentially surpass Melon if more species were introduced. In assessing relative abundance, we found that all dissimilarity metrics followed a similar trend, with the exception of Euclidean distance. When analyzing samples containing 32 species, Melon registered the least dissimilarity compared to all other tools. However, as complexity increased, Kraken began to take the lead. This shift was tied to the metric used: for Manhattan distance, Euclidean distance, as well as Bray–Curtis dissimilarity, Kraken emerged as the leading method when the species count approached approximately 64, whereas for Jensen–Shannon divergence, the transition occurred at a threshold of 512 species. It is also worth mentioning that the performance of Melon could further be boosted by increasing the length cutoff of its nucleotide database to 15,000 bp, yet the improvement from 10,000 bp to 15,000 bp was not as evident as from 5000 bp to 10,000 bp (Additional file 2: Fig. S2).

Lastly, we emphasize again that while DNA-to-DNA profilers (e.g., Kraken here) may demonstrate good performance in estimating both genome copies and relative abundances, their accuracy hinges on the comprehensiveness of the genome size database. In our simulation experiments, genome size information was available for all species, making it unsurprising that Kraken achieved commendable results. However, as many species in real-world metagenomes still lack complete genome representatives, deriving a

precise genome size database for correction can be very challenging. Melon, in contrast, does not face this limitation.

### Application to metagenomic samples

#### *Melon enables assembly-free, species-resolved ARG profiling in complex environmental metagenomes*

Long reads facilitate ARG host-tracking by providing contextual information about ARGs [45, 46]. This prompts us to ask whether it is possible to derive species-resolved ARG abundance estimates (in terms of ARG copies per cell) using Melon [47]. To this end, we constructed an ARG host database using the same strategy as before (Methods). ARG-containing reads were then mapped to this database for taxonomic identification. Note that we excluded all reads that mapped to mobile genetic elements (e.g., plasmids) since they are subject to horizontal gene transfer and may appear in multiple distant lineages [45, 48]. In a pilot test using mock sample S3 (D6300), we observed good congruence between estimated and expected species-level ARG abundances (Spearman's correlation $\rho = 0.915$, permutation test, two-sided, $p < 0.001$, Additional file 2: Fig. S3), despite some ARGs being underestimated due to identical cutoffs employed for both ONT reads and reference genomes (sequencing errors in ONT reads caused some boundary cases to be missed). Using 109 human fecal samples (sequenced by both Nanopore and Illumina) as an additional validation dataset [49], we observed a high correlation between the total ARG abundances estimated by this approach for long reads and ARGs-OAP [50] for short reads (Spearman's correlation $\rho = 0.941$, permutation test, two-sided, $p < 0.001$, Additional file 2: Note S3).

We then applied the method to two ONT samples: influent and effluent of a wastewater treatment plant located in Hong Kong (Shatin, 22.407° N, 114.214° E, Additional file 2: Table S1 and Table S3). Overall, we noticed that some ARG families were strongly phyla-specific. For example, rifamycin and mupirocin were found exclusively in Actinomycetota, whereas polymyxin, kasugamycin, florfenicol, and fosfomycin were only detected in Pseudomonadota (Fig. 4a). Other common ARG families, including macrolide-lincosamide-streptogramin (MLS), aminoglycoside, and tetracycline, were present in all four major phyla. In contrast, beta-lactam was absent in Actinomycetota, while vancomycin was missing in both Pseudomonadota and Bacteroidota.

Furthermore, wastewater treatments might alter the ARG abundances in specific species. For instance, the ARG abundance in *Enterobacter cloacae* increased after treatments, while the opposite trend was observed in *Klebsiella pneumoniae*. For the influent sample, 47.5% of classified ARG copies were associated with Pseudomonadota, followed by 26.7% with Bacillota. For the effluent sample, the distribution shifted: 41.3% were carried by Pseudomonadota and 34.4% by Actinomycetota. This shift was expected given that the relative abundances of ARG-carrying species changed drastically after treatments: Bacillota decreased from 0.063 to 0.016, whereas Actinomycetota increased from 0.038 to 0.147 (Fig. 4b).

#### *Species-level ARG abundances are enriched by wastewater treatments*

Although the overall ARG abundances decreased from 0.554 to 0.519, species-level ARG abundances were more inclined to rise after wastewater treatments (Fig. 4c). The median

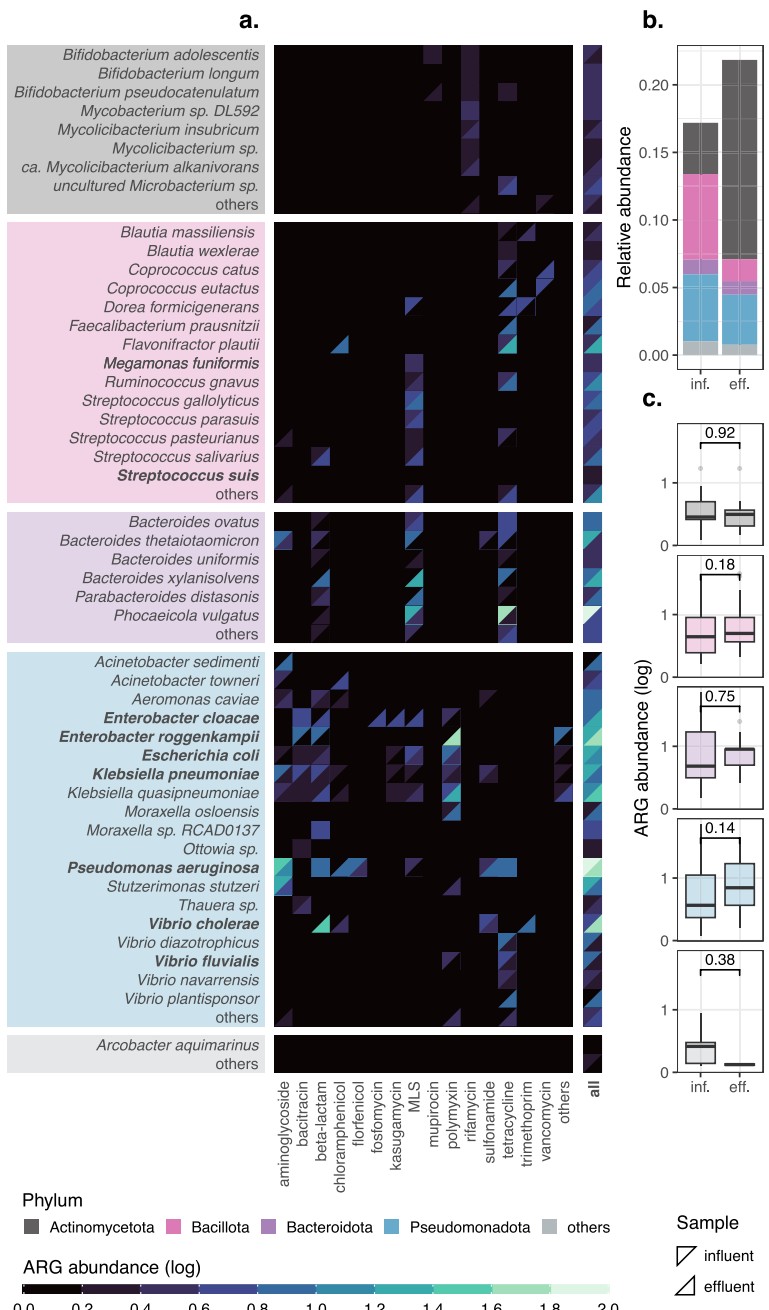

**Fig. 4** Application of Melon to influent and effluent samples. **a** Species-level ARG abundances of different ARG families for the influent (upper-left triangle) and effluent (bottom-right triangle) samples. Values are shifted by "+1" before log₁₀-transformation. Common bacterial pathogens (true or opportunistic), as classified by NCBI (https://www.ncbi.nlm.nih.gov/pathogens/organisms/), are marked in bold. Species not detected in either the influent or the effluent sample are labeled as "others." Only ARG-carrying species are displayed. **b** Relative abundances of ARG-carrying species, with unclassified species being removed. **c** Comparisons of ARG abundances in ARG-carrying species. Numbers indicate *p*-values returned by two-sided Wilcoxon rank-sum tests, without multiple testing correction. **b**–**c** Influent is abbreviated as inf. and effluent as eff

species-level ARG abundances for Actinomycetota increased from 1.8 to 2.1, for Bacillota from 3.4 to 4.0, for Bacteroidota from 4.0 to 8.0, and for Pseudomonadota from 2.7 to 6.0. Though not statistically significant (Wilcoxon test, two-sided), these findings,

combined with previous observations, suggest that the treatment process influences ARG patterns of wastewater samples, not just by changing their taxonomic composition, but also by enriching or depleting ARGs in specific species due to selective pressure. This also aligned with a previous study that indicated not all ARGs share the same fate during wastewater treatments [45].

## Discussion

Here we present Melon, a marker-based taxonomic profiler tailored specifically for error-prone long reads. Unlike sequencing-based methods, which typically require genome size correction to convert sequence abundance into taxonomic abundance, Melon naturally outputs species-level taxonomic abundance and genome copy estimates. Leveraging its EM-based post-correction module, Melon properly handles multimapped, ambiguous reads by reassigning them to the most likely lineages according to information collected from the entire sample. Using various sequenced and simulated samples, we show that Melon outperforms other existing tools in both species identification and quantification. We further illustrate the usage of Melon in real-world metagenomic studies with an example: influent and effluent from a wastewater treatment plant.

Melon is not the first tool that incorporates EM into long-read taxonomic profiling. MetaMaps is a DNA-to-DNA profiler that demonstrates good accuracy in taxonomic assignment by combining approximate read mapping with EM-based estimation of sample composition [27]. Although approximate read mapping excels in speed, it is not compatible with highly similar sequences, as noted by Emu [40]. Emu can be viewed as a quasi-DNA-to-marker profiler, as it targets amplicon sequences of the full-length 16S rRNA gene. Emu also employs minimap2's base-level alignment for mapping error-prone long reads and EM for resolving ambiguous mappings, thus has a high degree of similarity with Melon. However, both MetaMaps and Emu cannot yield taxonomic abundance in a straightforward fashion (genome size correction is required for MetaMaps; 16S copy number correction is required for Emu), which again highlights the importance of Melon.

In taxonomic profiling, marker genes should, at a minimum, meet the essential criteria of being both universal and single-copy. Ideally, these genes should also be highly conserved, yet effective in providing adequate taxonomic resolution when analyzed in tandem with adjacent genomic regions, and at the same time maintain a uniform or, at the very least, a balanced distribution across the genome to account for microbial growth. In this study, we focus exclusively on RPGs due to their high conservation. However, while they generally satisfy the essential criteria and suffice to differentiate species, selecting more than eight of them as marker genes for either bacteria or archaea is technically challenging because of their tendency to cluster closely on the genome. This relatively low number of marker genes makes Melon not well-suited for detecting ultra-low-abundance organisms, such as rare pathogens. Moreover, despite extensive efforts to select universal and single-copy marker genes, outliers do exist. For instance, *Cereibacter sphaeroides* from the ATCC 20 Strain Staggered Mix Genomic Material (MSA-1003) possesses two circular chromosomes and two exact copies of the bacterial RPG s7 and l11. Exceptional instances like this could distort both relative abundance and genome copy estimates, particularly if they are significantly represented in the community.

Incorporating additional marker genes or opting for lineage-specific marker genes—especially those other than RPGs—may offer advantages in both improving detection limits and counterbalancing the impact of outliers. We leave this for future research.

Another potential limitation of Melon pertains to its reference database. Unlike mOTUs, which excludes assemblies that fall below a certain marker gene count threshold, Melon incorporates all available assemblies into its database to maximize comprehensiveness. As genome databases like RefSeq and GTDB are rapidly expanding, they sometimes include draft assemblies that are either incomplete or contaminated. Such incompleteness could slightly inflate alpha-diversity estimates, as Melon attempts to identify closely related assemblies, which might not belong to the original species, in the absence of specific marker genes. Contamination could, on the other hand, result in erroneous species classification. However, the proportion of these anomalies is likely to be small and should not significantly impact Melon's overall performance, thanks to Melon's implementation of species-level clustering of databases (which addresses incompleteness) and EM-based post-correction of reads (which addresses contamination). Given that both NCBI and GTDB are actively engaged in maintaining database quality through tools like CheckM2, we delegate this quality control step to experts at RefSeq and GTDB.

Recent technological advancements have remarkably improved accuracy, read length, and throughput of long-read sequencing. Using accuracy of ONT reads as an example, the mean quality score has increased from approximately 10 (90% expected identity) in 2018 to 20 (99% expected identity) in 2023. These substantial improvements have motivated us to examine ONT samples generated through different combinations of sequencing chemistries and basecalling models, as characteristics of reads could be a critical factor that drives the performance of taxonomic profiling. Here, we focus primarily on ONT reads, as we empirically find that Pacific Biosciences (PacBio) HiFi reads can be seen as a special subset of recent ONT reads (less varied in read length, with near-perfect accuracy). It is not difficult to verify that Melon achieves equally good performance with PacBio HiFi mock samples such as ATCC MSA-1003 (excluding *Cereibacter sphaeroides*, as discussed earlier) and ZymoBIOMICS D6331 (Additional file 2: Table S3 and Fig. S4). Due to space constraints, we omit these two mocks from our main analyses.

In summary, given the growing interest in incorporating long reads into metagenomics, along with anticipated improvements in sequencing depth and more comprehensive databases, we believe that Melon will not only prove useful in taxonomic profiling but also pave the way for a wide range of applications, e.g., absolute quantification [51].

## Conclusions

Melon is a novel computational approach that aims to provide species-level taxonomic abundance profiles of complex microbial communities using long reads. Melon takes the advantages of long reads in their ability to offer higher taxonomic resolution and leverages EM to effectively handle the error-prone nature of these reads. Melon demonstrated better performance in terms of accuracy compared to other commonly used profiling tools, both on sequenced and simulated benchmarks. Melon can be installed via Bioconda and is available at https://github.com/xinehc/melon.

## Methods

### Database construction and Melon implementation

#### *Annotation of raw protein sequences*

Raw protein sequences (nr and env_nr, $n = 606,124,298$) were retrieved from the NCBI FTP server (https://ftp.ncbi.nlm.nih.gov/blast/db) on July 31, 2023. Bacterial and archaeal sequences ($n = 473,274,163$) were extracted from the nr database using TaxonKit v0.14.3 [52] and the 'blastdbcmd' command from BLAST+ v2.14.0 [53]. Sequences shared by both bacteria and archaea ($n = 67,318$) were discarded to avoid cross-mapping. For sequences of env_nr ($n = 10,951,228$), taxonomic information was not available. We thus mapped these sequences to the full nr database using DIAMOND v2.1.8 [17] ("blastp --top 0"), and subsequently removed every sequence whose best-scoring hits had a least common ancestor that was either unknown or non-prokaryotic ($n = 294,083$).

Remaining sequences ($n = 483,863,990$) were functionally annotated using the "hmmsearch" command from HMMER v3.3.2 [30] and the KOfam PHMMs database (ver. 2023-04-01). To reduce computational time, the annotation was carried out in two steps: we first extracted sequences that putatively contained at least one RPG using a subset of the KOfam database (91 PHMMs), and then annotated the selected sequences ($n = 3,927,293$) using the full KOfam database (11,298 PHMMs). The precomputed adaptive score thresholds associated with PHMMs were scaled by a factor of 0.75 to prevent dropping too many RPGs. To rule out spurious hits, sequences to which more than one PHMM was aligned (non-unique hits, $n = 38,568$) or whose length was covered by less than or equal to 75% by all aligned PHMMs (incomplete hits, $n = 199,811$), were discarded. With this, 3,688,914 sequences were classified as RPGs. To reduce redundancy, these sequences were further clustered using MMseqs2 v14.7e284 [16] ("easy-cluster -s 7.5 -c 0.98 --min-seq-id 0.98 --cov-mode 0 --cluster-reassign"), leading to a total number of 2,850,814 sequences.

#### *Evaluation of PHMMs*

The performance of the 91 RPG-related PHMMs was evaluated using the refseq_protein database. Briefly, we collected, extracted, and annotated sequences of prokaryotic refseq_protein using the same way as before. The PHMM-based annotations of the selected sequences were manually matched to their original RefSeq annotations by keywords. We considered RefSeq annotations as the "ground truth" and computed precision ($\frac{\text{TP}}{\text{TP+FP}}$), recall ($\frac{\text{TP}}{\text{TP+FN}}$), and $F_{0.5}$-scores ($\frac{1.25 \cdot \text{precision} \cdot \text{recall}}{0.25 \cdot \text{precision} + \text{recall}}$) for all available PHMMs. Here, TP stands for true positives, FP for false positives, and FN for false negatives. We chose refseq_protein for evaluation primarily owing to the fact that sequences in refseq_protein are annotated in a consistent manner and the annotations are in general of high-quality [8, 54]. Of note, despite extensive curation by NCBI, some annotations in the refseq_protein database might be incorrect. However, these erroneous annotations likely represent only a small fraction of the database and should not affect the conclusion regarding the overall performance of PHMMs.

### Selection of universal and single-copy marker genes

To assess whether RPGs met the criteria of being both universal and single-copy, we gathered 35,224 "complete genome" RefSeq assemblies from the NCBI FTP server (https://ftp.ncbi.nlm.nih.gov/genomes/refseq/). Completeness and contamination of these assemblies were predicted using CheckM2 v1.0.2 [36]. To address the overrepresentation of certain lineages (e.g., true or opportunistic pathogens), we retained only the best-scoring assembly for each species, with a score defined as completeness minus five times contamination [37]. Note that, unlike CheckM, CheckM2 is a machine-learning-based tool that does not rely solely on its predefined lineage-specific marker gene set (many of which are indeed RPGs) for genome quality assessment [36, 55]. As such, higher completeness predicted by CheckM2 does not necessarily indicate a more complete set of RPGs, and vice versa.

Assemblies were aligned to the clustered RPG database using DIAMOND's frame-shift alignment mode with an *e*-value cutoff of $10^{-15}$ and a subject cover cutoff of 75% ("blastx --evalue 1e-15 --subject-cover 75 --range-culling --frameshift 15 --range-cover 25 --max-target-seqs 25 --max-hsps 0"). We empirically find that this nucleotide-to-protein alignment is more straightforward and has better performance in comparison with the standard alignment pipeline (gene prediction followed by protein-to-protein alignment), as it not only properly handles potential frame-shifted genes induced by sequencing or assembly errors, but also mitigates the issue of misreported protein-coding genes (especially short genes and overlapping genes) arising from gene prediction [33, 56]. The resulting hits were filtered using in-house scripts. In brief, for each query sequence (contig), we sorted its hits by *e*-value in ascending order, and then iteratively added hits to a collection if and only if they exhibited less than 25% pairwise query range overlaps (computed as percentages of the shorter ranges) with any hits already in the collection. It is important to note that we allowed multiple copies of a specific gene sequence to be detected by setting '--max-hsp 0'.

We calculated for each available RPG its universality (defined as the proportion of assemblies containing it) and deviance (defined as the absolute difference between its mean copy and the ideal count of one) using (1) the whole collection of best-scoring assemblies ($n = 9195$) and (2) the tiny genome subset ($n = 234$). An RPG was considered valid if its universality exceeded 0.99, its deviance fell below 0.01, and its associated PHMM had a $F_{0.5}$-score above 0.99. Valid RPGs were hierarchically clustered (single-linkage) based on their mean relative genomic distances, calculated from all circular genome assemblies ($n = 8589$). Finally, we applied a distance cutoff of 0.05 and selected a single representative RPG from each cluster, according to universality and deviance, to serve as a marker gene. These marker genes (eight each for both bacteria and archaea) make up the protein database.

### Extraction of marker-gene-containing sequences

We retrieved 310,881 RefSeq assemblies from the NCBI FTP server (https://ftp.ncbi.nlm.nih.gov/genomes/refseq/) as of July 31, 2023. These assemblies were aligned against the protein database using the frame-shift alignment mode of DIAMOND (same configuration and post-filtering as before, see the "Selection of universal and single-copy marker genes" section). To minimize the inconsistency arising from partial alignments,

we discarded hits located close to the boundaries, specifically those with distances less than the lengths of their respective subject sequences (marker gene sequences). Additionally, we retained at most one hit per marker gene for each query sequence (contig), according to their alignment identity, to reduce the impact of misassemblies. For each of the filtered hits, a sequence of length 10,000 bp (left and right 5000 bp flanking regions of the hit's center) was extracted using SeqKit v2.5.1 [57] ("subseq"), whenever possible. The resulting marker-gene-containing sequences formed the initial nucleotide database. To further reduce redundancy, the database was clustered separately for each combination of marker genes and species at an identity cutoff of 0.9998 and a cover cutoff of 0.9998 (approximately one mismatch per 5000 bp) using MMseqs2 ("easy-cluster -s 7.5 -c 0.9998 --min-seq-id 0.9998 --cov-mode 1 --cluster-reassign"). The pairwise ANI of all marker-gene-containing sequences was computed using skani v0.2.1 [39] ("dist").

### *Implementation of Melon*

Melon takes quality-controlled long reads as input. In order to minimize the impact of potential unaddressed human contamination and/or non-prokaryotic sequences, we include an optional pre-filtering module built upon Kraken [58]. In short, we first predict the taxonomy of all reads using Kraken, and then eliminate any reads that belong to "Eukaryota," "Viruses," or "other entries." Of note, we chose Kraken mainly for its speed, but also acknowledge that Kraken's default fungi-covering databases (PlusPF, PlusPFP, etc.) contain only complete genome or chromosome level assemblies, so it might not be ideal if the sample contains many yet-to-be-characterized non-prokaryotic species. Other tools, such as EukRep, might be a viable alternative [59].

Quality-controlled and optionally decontaminated reads are aligned to the protein database using the frame-shift alignment mode of DIAMOND (same configuration and post-filtering as before, see the "Selection of universal and single-copy marker genes" section). For each detected subject sequence (marker gene sequence), we compute a trimmed mean of its coverage by slicing off the leftmost and rightmost 25% values. With a default subject cover cutoff of 75%, the trimmed mean here is exactly identical to the number of mapped hits. The coverage of each marker gene is then determined by summing up the coverage of all its constituent sequences. The total number of genome copies is estimated by taking the average coverage across all marker genes.

Reads containing at least one marker gene are mapped to the nucleotide database for taxonomic classification using minimap2 ("-cx map-ont -f 0 -N 2147483647 -p 0.9"). By default, we output all possible secondary alignments that exceed a secondary-to-primary score ratio of 0.9 during initial approximate mapping by setting "-N 2147483647 -p 0.9". To address the redundancy of the database, we disabled the option that ignores a fraction of the most frequent minimizers by setting "-f 0".

Alignments are sorted by alignment score (AS), local-best alignment score (MS), and gap-compressed per-base identity (ID) in descending order. For each read-species combination, only the best alignment is retained. Assuming reads are indexed by $i \in \{1, \ldots, n\}$ and species by $j \in \{1, \ldots, k\}$, an alignment is considered valid if it meets one of the following three conditions: (1) $\mathrm{AS}_{i,j} > 0.99 \cdot \max_j \mathrm{AS}_{i,j}$, (2) $\mathrm{MS}_{i,j} > 0.99 \cdot \max_j \mathrm{MS}_{i,j}$, or (3) $\mathrm{ID}_{i,j} > 0.999 \cdot \max_j \mathrm{ID}_{i,j}$. Next, we generate an $n \times k$ binary read-species matrix using

all valid alignments, with entries of the matrix representing the presence/absence of the corresponding read-species pairs. The matrix is subjected to EM for reassignment.

Let $\mathbf{x}_i = [x_{i,1}, \ldots, x_{i,k}]$ be a row of the above binary matrix where $x_{i,j} = 1$ if read $i$ maps to species $j$, and $x_{i,j} = 0$ otherwise. Let $\mathbf{z}_i = [z_{i,1}, \ldots, z_{i,k}]$ be a latent indicator vector where $z_{i,j} = 1$ if read $i$ truly originates from species $j$, and $z_{i,j} = 0$ otherwise. Note that, unlike $\mathbf{x}_i$ which satisfies $\sum_{j=1}^{k} x_{i,j} \geq 1$, $\mathbf{z}_i$ is a one-hot vector containing exactly one "1" and $k - 1$ "0"s, i.e., $\sum_{j=1}^{k} z_{i,j} = 1$. Additionally, we assume for each read one of the alignments is correct: if $z_{i,j} = 1$ then $x_{i,j} = 1$, but the reverse may not hold. With this notation, the relative abundances of species $j \in \{1, \ldots, k\}$ can be defined as $\boldsymbol{\theta} = [\theta_1, \ldots, \theta_k]$.

Our goal here is to find an estimate of $\mathbf{z}_i$ for all $i \in \{1, \ldots, n\}$. This objective can be achieved by iterating the E-step and M-step of the EM algorithm until convergence [60–62]. For simplicity, we use superscripts to represent the number of iterations and initialize $\theta_j^{(0)} = \frac{1}{k}$ for all $j \in \{1, \ldots, k\}$. In the E-step, we find the soft latent assignment $z_{i,j}$ by setting it to the posterior probability, given the current estimate of $\theta_j$:

$$z_{i,j}^{(t+1)} = \frac{x_{i,j}\theta_j^{(t)}}{\sum_{j=1}^{k} x_{i,j}\theta_j^{(t)}}, \ \forall i, j.$$

In the M-step, we derive a new estimate of $\theta_j$ by maximizing the expected complete data log-likelihood:

$$\theta_j^{(t+1)} = \frac{\sum_{i=1}^{n} z_{i,j}^{(t+1)}}{n}, \ \forall j.$$

These two steps are iterated until one of the following two conditions is satisfied: (1) $\sum_{j=1}^{k} | \theta_j^{(t+1)} - \theta_j^{(t)} | < \varepsilon$ or (2) $t > t_{\max}$, with default values being set to $\varepsilon = 10^{-5}$ and $t_{\max} = 100$. Once the iteration stops, the most likely species associated with each individual read is determined using a hard assignment, specifically, $\arg\max_j z_{ij}$. In cases where there is a tie in the values of $\mathbf{z}_i$, we consider additional metrics such as AS, MS, and ID. We also evaluate whether the target species have been properly defined, for example, *Escherichia* sp. is weighted less than *Escherichia coli*. Finally, we aggregate the taxonomic information of all reads, resulting in a tab-delimited table listing species and their copy numbers.

### Analysis of replicating species with circular genomes

Short and long reads of *Bacillus amyloliquefaciens* (Gram-positive) and *Pseudomonas alloputida* (Gram-negative), were collected from a previous study [51]. Short reads were quality-controlled with fastp v0.23.4 [63], while long reads were processed with Porechop v0.2.4 [64] ("--discard_middle") and nanoq v0.10.0 [65] ("--min-qual 10 --min-len 1000"). Cleaned reads were assembled using the hybrid mode of unicycler v0.5.0 [66]. We next mapped short reads back to the assembled circular genomes using minimap2 v2.26 ("-ax sr --secondary=no") and computed per-base coverage at all genomic locations using samtools v1.17 [67] ("sort" and "depth -J -a"). Per-base coverage was smoothed using a 5000-bp non-overlapping window where necessary. Regarding marker genes, we collected four different sets: Melon (8 genes), mOTUs [21] (10 genes), MicrobeCensus [68] (30 genes), and GTDB [35] (120 genes). The locations of these marker

genes were predicted using the frame-shift alignment mode of DIAMOND (same configuration and post-filtering as before, but with "--id 75" for removal of spurious hits, see the "Selection of universal and single-copy marker genes" section). The PTRs of these two replicating species were estimated using short reads and CoPTR v1.1.2 [69] with default parameters.

## Performance evaluation

### Collection of mock samples

One of the D6300 (S2) and two of the D6331 (G1–2) mock samples were collected from previous studies [70, 71]. The remaining two D6300 mock samples (S1 and S3) were downloaded from an online source (https://lomanlab.github.io/mockcommunity/r10.html). The last D6331 mock sample (G3) was sequenced in our lab using the latest ONT Q20+ kit (V14 kit chemistry and R10.4.1 pore). Briefly, DNAs were extracted from ZymoBIOMICS Gut Microbiome Standard using DNeasy PowerSoil Pro Kit (Qiagen) according to the manufacturer's protocol. Quality and concentration of the extracted DNAs were then measured by NanoDrop One (Thermo Fisher Scientific) and Qubit 2.0 Fluorometer (Thermo Fisher Scientific), respectively. Library preparation was carried out using SQK-NBD114 native barcoding kits for sequencing on R10.4.1 flowcells, following the manufacturer's protocol. Reads were generated on PromethION using the MinKNOW v23.04.5 software with Guppy v6.5.7 (400 bps, 5 kHz, super-accurate, https://community.nanoporetech.com/).

All samples were quality-controlled using Porechop ("--discard_middle") and nanoq ("--min-qual 10 --min-len 1000"). We then mapped these samples back to their reference genomes using minimap2 ("-ax map-ont --secondary=no"). All reads that aligned to yeasts or remained unaligned were discarded using customized scripts. Per-base coverage at all genomic locations was determined for each species using samtools ("sort" and "depth -J -a"). To avoid the influence of multi-copy plasmids, we considered the mean coverage of each species' longest contig as its expected genome copy. The total genome copy of a sample was obtained by summing up the genome copies of all its species.

For each of the mock samples, we trained a model using the "read_analysis" script ("metagenome") from NanoSim v3.1.0 [44]. This resulted in six models, each with a different error profile.

### Comparison of tools

The performance of Melon v0.1.0, Kraken v2.1.3 [11], mOTUs v3.1.0 [21], and Bracken v2.8 [10] was evaluated. Since Kraken and Bracken require genome size information to convert sequence abundance into taxonomic abundance, we constructed a hierarchical genome size database using all RefSeq complete genome assemblies ($n = 35,224$). In brief, we kept track of all available NCBI taxids (taxonomy IDs) and, for each taxid, estimated its genome size by averaging the sizes of all its descendant genomes. For simplicity, we assumed all plasmids and other extrachromosomal genetic elements were of single-copy, and included them in both the estimation of genome sizes and the simulation experiment (see the "Simulation experiment" section). With this, we obtained a genome copy estimate for each species by dividing the number of base pairs assigned

to it by its corresponding genome size. In instances where the genome size of a species was unknown, the genome size of its lowest taxonomic ancestor was used for correction, whenever possible. For Bracken, since taxonomic assignments of individual reads are not given, we assumed all reads had a length equal to the sample's average read length. Melon was run using default parameters and the latest NCBI database (ver. 2023-07-31). For Kraken and Bracken, the Standard database (ver. 2023-06-05, downloaded from https://benlangmead.github.io/aws-indexes/k2) was utilized. Kraken was executed with default settings, while Bracken was configured to output species-level results by setting "-l S -r 50". For mOTUs, we set it to output taxids and base coverage using "-p -c -y base.coverage". All taxonomic assignments above the level of species were marked as unclassified.

Regarding evaluation metrics, $x/y$ ratio is defined as the ratio between the estimated and expected total genome copies. Pearson's correlation measures the linear relationship between the estimated and expected total genome copies and is defined as the ratio between the covariance of these two and the product of their standard deviations. Precision is again defined as $\frac{\text{TP}}{\text{TP+FP}}$ and recall as $\frac{\text{TP}}{\text{TP+FN}}$, but here the meaning of true positives is the number of species observed in both the estimated and expected profiles. $F_1$-score is the harmonic mean of precision and recall, defined as $\frac{\text{precision·recall}}{\text{precision+recall}}$. The four dissimilarity/distance metrics are defined elsewhere and were computed using SciPy v1.10.1 [72] based on the estimated and expected relative abundances. All metrics, except for Pearson's correlation, were computed separately for each sample and aggregated using arithmetic means, where necessary. To ensure a fair comparison, we excluded all unclassified species when calculating precision, recall, $F_1$-scores, and all dissimilarity/distance metrics.

### Harmonization of databases

Since the performance of taxonomic profilers is profoundly influenced by their reference databases, benchmarking without considering the inconsistency of databases may result in unfair comparisons and biased conclusions [6]. In light of this, we aimed to harmonize the databases of Melon, Kraken (Bracken), and mOTUs by identifying their common species prior to simulation.

In essence, given a list of candidate assemblies, we first examined their presence in the databases of Melon and Kraken. Melon's NCBI database can be viewed as a "superset" of Kraken's Standard database (prokaryotic part), though there are exceptions like deprecated assemblies that are no longer available from NCBI. Assemblies present in both Melon's and Kraken's databases were then incorporated into the database of mOUTs using its auxiliary tool, "mOTUs-extender" (https://github.com/motu-tool/mOTUs-extender). Assemblies that failed to be added due to insufficient marker genes were subsequently discarded. Through this approach, we ensured a consistent presence of species across all databases.

### Simulation experiment

The 64 mouse gut profiles were downloaded from the second CAMI challenge (https://repository.publisso.de/resource/frl:6421672). The remaining $5 \times 10$ metagenomic profiles were generated by randomly sampling 32–512 species, without replacement, from

a pool of 8618 species and assigning each species a taxonomic abundance according to a log-normal distribution ($\mu = 0$, $\sigma = 1$). For each of these profiles, we simulated six samples (500,000 reads each) using previously described models (see the "Collection of mock samples" section) and the "simulator" script ("metagenome") from NanoSim.

Since NanoSim does not accept taxonomic abundance as input, we converted species' taxonomic abundances into sequence abundances according to the lengths of their contigs ("--abun"). Additionally, we enabled the simulation of circular genetic elements by explicitly specifying their topologies in the input files ("--dna_type_list"). The expected genome copy of each individual species was calculated by counting the number of base pairs aligning to its longest contig, as indicated in the headers of the output files, and subsequently dividing this count by the contig's length. We assessed the performance of different tools on these simulated samples using the same approach as before (see the "Comparison of tools" section)

### Application to metagenomic samples
#### *Collection of wastewater samples*
The influent and effluent samples were collected from the Shatin wastewater treatment plant (22.407° N, 114.214° E) on June 12, 2023. These two samples were prepared and sequenced using the same strategy as described earlier (see the "Collection of mock samples" section), except that they were spiked with two external species, *Allobacillus halotolerans* and *Imtechella halotolerans*, for other purposes. For consistency, reads of these two species were removed using minimap2 ("-ax map-ont --secondary=no") at a cutoff of 90% gap-compressed per-base identity. All samples underwent quality control using Porechop ("--discard_middle") and nanoq ("--min-qual 10 --min-len 1000").

#### *Estimation of genome copies, mean genome sizes and ARG abundances*
Genome copies of individual species were predicted using Melon (pre-filtered with Kraken's PlusPF database, ver. 2023-06-05). Mean genome sizes of prokaryotes were estimated by dividing the total number of base pairs (excluding contributions from human and other non-prokaryotes) by the total number of genome copies. ARG copies were determined by mapping filtered reads to the SARG v3.2.1-F [50] database using the frame-shift alignment mode of DIAMOND (same configuration and post-filtering as before, but with "--id 75" for removal of spurious hits, see the "Selection of universal and single-copy marker genes" section). Hits were excluded if their predicted ARG families were flagged as multidrug, since multidrug efflux pumps have a broad range of substrates that are not necessarily specific to antibiotics [51]. ARG abundances, expressed as "ARG copies per cell" (assuming one genome copy per cell), were calculated by dividing the estimated ARG copies by the total genome copies. For Illumina samples, ARG abundances were provided by ARGs-OAP [50] v3.2.4.

#### *Identification of ARG hosts*
We constructed an ARG host database using the same approach as before (see the "Extraction of marker-gene-containing sequences" section). However, while the nucleotide database kept at most one hit per marker gene for each contig, here we maintained all detected hits, regardless of their ARG families. We then obtained the

sources (chromosomes, plasmids, and viruses) of these sequences using geNomad v1.7.0 [73] ("end-to-end") with default parameters.

ARG-containing reads were mapped to the ARG host database and filtered using the same strategy as previously described (see the "Implementation of Melon" section). However, an exception was made in the filtering step: we removed alignments if their predicted species were not present in Melon's results and retained only a single best alignment for each read. After this step, we omitted all reads that mapped to plasmids or viruses to mitigate the influence of mobile genetic elements. For each species, we counted its ARG copy from the filtered alignments and then determined its ARG abundance by dividing the number of ARG copies by the number of genome copies.

### *Statistical analysis*

All statistical analyses were conducted in R v4.3.1 [74]. Wilcoxon rank-sum test was performed with the "wilcox.test" function. Permutation test for Spearman's correlation was done using customized scripts, with 9999 permutations. All figures were generated using "ggplot" v3.4.3 [75]. Results were deemed statistically significant if $p < 0.05$. No multiple testing correction was done unless otherwise stated.

## Supplementary Information

> Additional file 1. Supplementary Data S1–3.
>
> Additional file 2. Supplementary Tables S1–3, Supplementary Figures S1–4, Supplementary Notes S1–3.
>
> Additional file 3. Review history.

### Acknowledgements
X.C., X.S., Y.Y., and W.Y. would like to thank the University of Hong Kong for the postgraduate studentship. X.Y. and L.L. would like to thank the University of Hong Kong for the postdoctoral fellowship. The computations were performed using research computing facilities offered by Information Technology Services, the University of Hong Kong.

### Review history
The review history is available as Additional file 3.

### Peer review information
Kevin Pang and Andrew Cosgrove were the primary editors of this article and managed its editorial process and peer review in collaboration with the rest of the editorial team.

### Authors' contributions
X.C., X.Y., and T.Z. conceived this study. X.S., Y.Y., and L.L. sequenced the mock and wastewater samples. W.Y. collected the wastewater samples, X.C. developed the tool and analyzed the data. X.C. and T.Z. interpreted the results. X.C. wrote the original draft of the manuscript. All authors contributed to the revision of the manuscript.

### Funding
This study was financially supported by the Theme-based Research Scheme (T21-705/20-N) of Hong Kong.

### Availability of data and materials
Sequenced samples (mock sample G3, influent, and effluent) are uploaded to NCBI Sequence Read Achieve (SRA) under BioProject ID PRJNA1028177. Mock samples S2 and G1–2 are available under PRJEB29504 and PRJNA804004, respectively. Mock samples S1 and S3 can be downloaded from https://lomanlab.github.io/mockcommunity/r10.html. PacBio mock samples are collected from https://github.com/PacificBiosciences/pb-metagenomics-tools/blob/master/docs/PacBio-Data.md. Short and long reads of the two replicating species can be found under PRJNA895741. The human fecal dataset is available under PRJEB49168. The profiles used for simulation are deposited at https://doi.org/10.5281/zenodo.12770347 [76]. Melon is available under the MIT License at GitHub (https://github.com/xinehc/melon) [77], and can be installed via Bioconda (https://anaconda.org/bioconda/melon). The source code of Melon v0.1.0 is available at https://doi.org/10.5281/zenodo.12770296 [78]. Detailed instructions on database construction can be found in the supplementary repository at GitHub (https://github.com/xinehc/melon-supplementary) [79].

## Declarations

**Ethics approval and consent to participate**
Not applicable.

**Consent for publication**
Not applicable.

**Competing interests**
The authors declare that they have no competing interests.

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

## 