## [Additional file 3. Review history. · Genome Biology]

1st round

Reviewer 1

In this study, the authors bridge a significant gap between metagenomic sequencing and data processing by developing 'Melon,' a novel marker-based taxonomic profiler tailored for long-read sequencing data. This innovative tool represents a significant advancement over existing methodologies, such as mOTUs3, Kraken2, and Bracken, particularly in its native ability to process long-read sequences without the need for preprocessing steps like fragmenting into short reads. The comprehensive evaluation conducted by the authors, including both in vitro mock and in silico simulated samples, showcases Melon's superior performance in species identification and abundance estimation. The paper is well-written and employs appropriate methods for simulation and evaluation. It will undoubtedly capture the interest of researchers in microbiome and environmental microbiology fields, offering a valuable tool for the accurate and efficient analysis of long-read sequences. Moreover, the significance of this work extends to public health, particularly in its potential application for tracking antibiotic resistance genes (ARGs) in complex environments such as wastewater. Before proceeding, I would recommend the authors address the following questions.

Line 38: "marker" in the so-called marker-based profiler still refers to a sequence, so I suggest retaining the original terminology from the paper to avoid confusion, such as using DNA-to-DNA/Protein or DNA-to-Marker method. Please ensure this consistency is maintained throughout the manuscript.

Line 76: why is this approach considered inappropriate, and how does it impact the results? The manuscript would benefit from a more detailed explanation.

Line 112: if F0.5 is the weighted harmonic mean of precision and recall, this should be explained at its first mention, along with clarifications on F1 and x/y ratios.

Line 123: the feasibility of aligning 0.9 million sequences on a standard laptop is unclear without benchmarking for the computing resources required. Although this represents a significant reduction in the number of reads, it's still a substantial volume. How large is the database?

Line 133: using only one representative genome for assessment raises concerns about potential bias. For instance, how can you be sure that the other qualified genomes of *E. coli* don't vary in RPG copy number? This aspect needs addressing to validate the evaluation's robustness.

Line 163, I believe this is critical in determining the resolution of Melon. Please add the distribution of the similarity of selected markers among different species/strains, which will give readers a clearer picture of the accuracy and potential of these marker genes.

I am very interested to see how the RPGs assessment goes in GTDB instead of RefSeq,

considering that RefSeq has issues with confused taxonomy annotation. Since GTDB relies solely on genome similarity, it could potentially retain more markers, possibly enhancing abundance estimation with a larger pool of marker genes.

Line 231: the explanation of Bracken and mOTUs' performance is confusing. Please provide a clearer explanation and define what is meant by the recovery rate.

Line 237: what was the rationale behind choosing 500,000 reads for the simulation??

Line 254: in which species richness? or are you using the Wilcoxon signed-rank test?

Line 300: It's not just Kraken that faces the problem mentioned; it's a common issue for every metagenomic profiler. The discussion could instead focus on the type of relative abundance generated by DNA-to-DNA methods.

Line 306: it is good to see the application of Melon, but adding some validation, such as using shotgun sequencing data and a well-established pipeline for ARG calculation as a comparison, would greatly enhance its credibility.

Additionally, given that the primary difference between Melon and mOTUs is the selection of marker genes and the alignment of flanking sequences used in abundance estimation, I see no reason why this method couldn't also be applied to short sequencing reads. Perhaps this could be evaluated in the manuscript or considered for future work.

Reviewer 2

The authors presented a new tool Melon for long-read based taxonomic profiling. The authors justified the usage of marker-gene based method and tested Melon in both mock and real data. My main concern is that while the authors described Melon as a profiler tailored specifically for error-prone long reads, the design of the methodology and the benchmarking experiments did not incorporate the properties of TGS or the tools that are specifically designed for TGS. Thus, it is not clear whether the main contribution is the set-up of the reference maker database or the pipeline.

Major:

As there are previous research about single copy marker genes, have the authors compared their markers with those single copy genes? Seems using more single-copy marker genes is expected to achieve a higher recall while simplifying the abundance computational.

As long reads can contain insertion/deletion sequencing errors, I am wondering whether the authors did any error correction, which can affect the gene prediction and translation. Or is DIAMOND under some setting is very robust against sequencing errors? Did the authors consider using optimized tools for long read alignment?

Until Section 5.1.5, it is not obvious which part of the methodology design is optimized for

long reads. In the experiments, the reviewer also expects to see the performance of Melon w.r.t. some data properties of TGS. For example, how does Melon's performance change with the read length, error rate, and coverage? Many TGS data may not be able to achieve a very high coverage. Seems these key factors are not evaluated.

How important is the EM step? Seems its contribution is not evaluated in the Results Section. It is only first mentioned in Discussion.

The chosen benchmark tools are not designed for long reads. Kraken is not specifically designed for long reads. Also, as the authors pointed out, BWA in mOTU is also not optimized for long reads. Thus, the comparison is not very fair. Although the authors mentioned that MetaMaps, MEGAN-LR, and BugSeq are "sequence-based", can they be customized with different reference database? If so, you can replace their reference data with your protein database? Additionally, you can also directly run Melon against these tools and compare their final outputs. A minor comment is related to the names of the two methods. After all, marker genes are still sequences. So using "sequence-based" to exclude marker-based tools can be confusing. Maybe "genome-based" is a better name.

Method 5.1.1 It will be clearer to describe the change of the number of sequences w.r.t. the multiple "filtration" strategies. In particular, it is not clear what is the purpose of the second alignment with full KOfam database. Also, is the scaled cutoff applied to both the 91 and 11,298 pHMMs or just the latter? The described procedure is hard to reproduce.

Overall, I found the descriptions at several places are hard to follow partly because of the scattered description of the major steps in different places and partly because the description is like a laundry list lacking justification. Two examples are given below. The second paragraph of Section 2.1 is a little hard to follow. The authors said "four factors are considered". But there lacks clear description of how to reduce 91 pHMMs to 8 pHMMs and how each factor is used specifically. Later, seems '8' refers to only 8 marker gene not 8 gene clusters (pHMMs) in 5.1.3. Another example. Up to Page 10, it is still not clear how the authors identify reads that cover marker genes. The authors mentioned "Melon first extracts reads that cover at least one marker gene using a protein database". Fig. 1 said "Aligning sequences of assemblies to the marker gene database". Neither of "extract" nor "align" is very specific. In 5.1.5, the authors mentioned the usage of "DIAMOND" for aligning RefSeq assemblies with the protein database. Better highlight the used method/tool for the major steps when those steps are introduced.

There are a large number of cutoffs/parameters. Is there a better way to present them and highlight the justifications?

Minor:

Does the last sentence mean that the authors applied the 8 pHMMs to all nr and env-nr proteins and kept the aligned proteins? Reading the detailed methods can help understand these

descriptions. But reading the paper following the given order can be a little confusing.

Melon is a computational tool. Thus, Fig. 1 a. is a little confusing as it contains sampling, DNA extraction, sequencing.

2.3.2 please specify the tool you used to simulate TGS data at its first mention.

Reply to the review report on

GBIO-D-23-01745: Melon: metagenomic long-read-based taxonomic identification and quantification using marker genes

June 4, 2024

Dear Editor,

Thank you for your time and effort in handling our manuscript. Enclosed please find a
point-by-point response to the reviewers' comments and the revised manuscript entitled
"**Melon: metagenomic long-read-based taxonomic identification and quantification**
**using marker genes**" for consideration of publication in *Genome Biology*.

We thank the reviewers for their many insightful comments and constructive suggestions,
which have helped us improve and clarify the manuscript. We also appreciate the opportunity
to submit the revision and would like to highlight the following major changes:

- • We expanded the evaluation of marker genes by including average nucleotide identity
comparisons of marker-gene-containing sequences at strain or species level.
- • We compared the results of using GTDB or RefSeq for marker-gene selection.
- • We added a validation dataset comprising 109 human fecal samples to justify the
application of **Melon** in antibiotic resistance gene quantification.
- • We included a benchmark experiment to demonstrate **Melon**'s advantage over existing
long-read-specific profiles, including **MetaMaps** and **MEGAN-LR**.
- • We showed the robustness of **Melon** with respect to the properties of third-generation
sequencing (TGS), including quality score q and read length l , using a mock sample.
- • We revised Fig. 1 and Sections 2 & 5 to enhance the overall readability.

In the point-by-point response, reviewers' comments are marked with red box, followed by
our responses in plain black and changes in blue box. In the revised manuscript, changes
are tracked with **red** or blue text.

Sincerely,
Tong Zhang

**Reviewer 1**

In this study, the authors bridge a significant gap between metagenomic sequencing and
data processing by developing ‘Melon’, a novel marker-based taxonomic profiler tailored
for long-read sequencing data. This innovative tool represents a significant advancement
over existing methodologies, such as mOTUs3, Kraken2, and Bracken, particularly in its
native ability to process long-read sequences without the need for preprocessing steps like
fragmenting into short reads. The comprehensive evaluation conducted by the authors,
including both in vitro mock and in silico simulated samples, showcases Melon’s superior
performance in species identification and abundance estimation. The paper is well-written
and employs appropriate methods for simulation and evaluation. It will undoubtedly capture
the interest of researchers in microbiome and environmental microbiology fields, offering a
valuable tool for the accurate and efficient analysis of long-read sequences. Moreover, the
significance of this work extends to public health, particularly in its potential application for
tracking antibiotic resistance genes (ARGs) in complex environments such as wastewater.
Before proceeding, I would recommend the authors address the following questions.

**Author response:**

Thank you for the comments. We have expanded the evaluation of marker genes by including
average nucleotide identity comparisons and have used a human fecal dataset to demonstrate
the validity of ARG quantification. Please find our point-by-point responses below.

**Comment 1**

Line 38: “marker” in the so-called marker-based profiler still refers to a sequence, so I suggest retaining the original terminology from the paper to avoid confusion, such as using DNA-to-DNA/Protein or DNA-to-Marker method. Please ensure this consistency is maintained throughout the manuscript.

**Reply:**

We agree that using *sequence-based* and *marker-based* together can lead to confusion. We
have changed all *sequence-based* to *DNA-to-DNA/protein*, and most *marker-based* to *DNA-*
*to-marker*, following the notation given by Sun et al., 2021 and Ye et al., 2019.

**Changes:**

Here is one example of the changes in Section 1:

Depending on the type of the database employed, these classification methods can be broadly categorized into two groups: (1) ~~sequence-based~~ DNA-to-DNA/protein and (2) ~~marker-based~~ DNA-to-marker taxonomic profilers (Sun et al., 2021; Ye et al., 2019).

**Comment 2**

Line 76: why is this approach considered inappropriate, and how does it impact the results? The manuscript would benefit from a more detailed explanation.

**Reply:**

Cutting long reads into artificial short reads removes all long-range information (Portik
 et al., 2022). Given that most marker genes are longer than 300 bp, these short reads
 may not be able to cover the full range of the genes, leading to ambiguous alignments
 and, as a consequence, lower taxonomic resolution. In Fig. 3a, we see that mOTUs clearly
 underestimated the number of total genome copies for the mock samples (S1-3 and G1-3).
 Although this large discrepancy might be caused by the inappropriate choice of alignment
 settings and tools (i.e., BWA with presets for Illumina reads), we noticed that, on average
 5.35% of the estimated copies were not assigned with species-level taxonomic IDs. Therefore,
 we believe this strategy of cutting long reads into short reads is more of a compromise and
 not well-suited for long-read metagenomics.

Figure 3a: Performance of Melon on mock samples. Mock experiment (mock samples S1-3 and G1-3). Colors indicate tools used for comparison, including Melon (v0.1.0, NCBI database, ver. 2023-07-31), Kraken (v2.1.3, Standard database, ver. 2023-06-05), Bracken (v2.8, same database as Kraken) and mOTUs (v3.1.0, default database, with modifications). Shapes of points denote different mock communities, D6300 and D6331. Tables provide summarized metrics, including: ratio (x/y ratio), corr. (Pearson’s correlation), pr. (precision), re. (recall), and F₁ (F₁-score). Abbreviations used for tools’ names are: M. (Melon), K. (Kraken), m. (mOTUs), and B. (Bracken). Error bars depict the ranges of possible values. All metrics, except for Pearson’s correlation, are aggregated by averaging where necessary.

**Changes:**

We have added a line to clarify the limitation of cutting long reads into short reads:

Although mOTUs3 recently introduced a preprocessing step ~~which that~~ enables it to handle long reads by fragmenting ~~long reads them~~ into short reads, it cannot be regarded as a native long-read method due to the loss of long-range information (Ruscheweyh et al., 2022; Portik et al., 2022).

**Comment 3**

Line 112: if $F_{0.5}$ is the weighted harmonic mean of precision and recall, this should be explained at its first mention, along with clarifications on F_1 and x/y ratios.

**Changes:**

The definition of $F_{0.5}$ -scores has now been added in Section 2.1:

A subset of RPGs were screened as marker genes (eight each for both bacteria and archaea, see section “Quality assessment of PHMMs and RPGs” for more details) by assessing the quality of each according to four distinct factors: (1) $F_{0.5}$ -scores ~~of~~ (weighted harmonic mean of precision and recall) of their associated PHMMs, (2) their prevalence among species, (3) discrepancies between their average numbers of copies and the ideal count of one, and (4) their mean relative genomic distances to other candidate RPGs (Fig. 1c).

We have added clarifications to F_1 -scores and x/y ratios in Section 2.3.1:

However, as reflected by their inferior x/y ratios ~~and~~ (ratios between the estimated and expected total genome copies, i.e. recovery rate of genome copies) and Pearson’s correlations, the estimates of Kraken and Bracken tended to fluctuate around the expected values.

mOTUs, on the other hand, was capable of providing balanced precision and recall, yet its overall performance, including F_1 -score (unweighted harmonic mean of precision and recall), was still lower in comparison to Melon.

**Comment 4**

Line 123: the feasibility of aligning 0.9 million sequences on a standard laptop is unclear without benchmarking for the computing resources required. Although this represents a significant reduction in the number of reads, it’s still a substantial volume. How large is the database?

**Reply:**

We recorded the run time and peak memory usage of **MelOn** on the influent and effluent
 samples (sequenced with Nanopore R10.4.1). For a 5 Gb effluent sample, **MelOn** was able to
 finished in 23 minutes on an M1 Max chip MacBook Pro (profiling against 310,881 RefSeq
 assemblies). The full resource requirements are shown in Supplementary Table S1.

The database after compression is roughly 2.9 GB for RefSeq R219 ([https://zenodo.org/](https://zenodo.org/records/8418111)
 [records/8418111](https://zenodo.org/records/8418111)) and 4.3 GB for GTDB R214 (<https://zenodo.org/records/8418135>).

Table S1: Computational time and peak memory usage for wastewater samples

		no pre-filter	PlusPF-8	PlusPF-16	PlusPF ^b
influent 7.816 Gb	genome copy	1,809	1,801	1,800	1,797
	species richness	2,101	2,098	2,099	2,097
	number of filtered reads	-	8,312	10,212	14,988
	mean genome size (Mb)	4.322	4.304	4.300	4.292
	ARG abundance (copies per cell) ^a	0.551	0.553	0.553	0.554
	real time (sec) ^c	2,056 967	2,271 1,027	2,279 1,151	- 1,272
	peak resident set size (GB) ^c	10.629 17.860	10.665 18.143	17.294 17.986	- 77.694
effluent 5.158 Gb	genome copy	1,348	1,336	1,331	1,315
	species richness	1,704	1,697	1,700	1,696
	number of filtered reads	-	16,602	29,300	54,774
	mean genome size (Mb)	3.826	3.789	3.757	3.715
	ARG abundance (copies per cell) ^a	0.507	0.511	0.512	0.519
	real time (sec) ^c	1,341 671	1,496 885	1,495 867	- 929
	peak resident set size (GB) ^c	10.893 13.649	10.902 13.227	17.067 17.454	- 77.325

^a Excluding multidrug ARGs.

^b Not tested with MacBook Pro due to insufficient memory.

^c Excluding ARG abundance estimation, measured with GNU ‘time’. **Red:** MacBook Pro 2021, with Apple M1 Max, 64 GB memory, and macOS Sonoma 14.0. **Black:** Lab-scale workstation, with 2 × Intel Xeon Silver 4210R CPU 2.40GHz (10 cores, 20 threads), 512 GB memory, and Ubuntu 20.04 LTS.

**Comment 5**

Line 133: using only one representative genome for assessment raises concerns about potential bias. For instance, how can you be sure that the other qualified genomes of *E. coli* don’t vary in RPG copy number? This aspect needs addressing to validate the evaluation’s robustness.

**Reply:**

The main reason we chose a single representative genome for each species was to avoid the
 overrepresentation of model/pathogenic species. For instance, *E. coli* accounted for around
 10% of all RefSeq complete genome assemblies (2,834 out of 35,224). Considering all complete
 genome assemblies of *E. coli* might be unfair to other underrepresented rare species and
 could lead to over-optimistic results.

The copy number of marker genes indeed varies across assemblies of species, yet at least for
 *E. coli*, this variation is more likely caused by the incompleteness of assemblies. For instance,
 99.4% of *complete genome E. coli* assemblies had exactly eight marker genes, but this number
 dropped quickly to only 14.6% for *contig* level assemblies (see Fig. R1 below).

Figure R1 for reviewer 1: Marker gene distribution of 34,126 *E. coli* assemblies.

In Fig. 2d, we see that almost all RefSeq complete genome assemblies possessed exactly eight
 marker genes. Moreover, for both bacteria and archaea, assemblies with more than eight
 marker genes tended to have high contamination, whereas assemblies with fewer than eight
 marker genes low completeness (predicted by CheckM2).

Figure 2d: Selection of marker genes. Numbers of marker genes detected in RefSeq assemblies. Colors indicate assembly levels. Completeness (cp.), contamination (ct.), and their relations to the number of marker genes are shown in tables.

**Comment 6**

Line 163, I believe this is critical in determining the resolution of Melon. Please add the distribution of the similarity of selected markers among different species/strains, which will give readers a clearer picture of the accuracy and potential of these marker genes.

**Reply:**

We thank the reviewer for pointing this out. The within-species/between-strains similarity of
marker-gene-containing sequences is indeed crucial in determining the taxonomic resolution
of **Melon**. This was overlooked in our previous assessments of marker genes. We have now
included a new experiment to compare the similarity of marker-gene-containing sequences at
strain or species levels.

Briefly, we computed the pairwise average nucleotide identity (ANI) between all marker-gene-
containing sequences (at most 10,000 bp) using `skani v0.2.1 (skani dist)` (Shaw et al.,
2023). The identity of a pairwise comparison was recorded if the aligned fraction was greater
than 75% for both the query and the reference.

Figure S5: ANI between marker-gene-containing sequences. Colors indicate within-species/within-genus identity. Dashed lines represent the 95% ANI cutoff for species identification. Archaeal ribosomal protein genes s2, s7, and s9 were not included.

As shown in Fig. S5, the median ANI ranged from 99.0% (bacterial s16) to 99.8% (bacterial
l2) between strains and from 93.8% (archaeal l2) to 96.9% (archaeal l10e) between species.
Given that RPGs are in general more conserved compared to other nonessential genes, it was

expected that some between-species ANIs exceeded the conventional ANI cutoff for species
identification, i.e., 95%. However, for most marker genes, there was still a gap between
the within-species and within-genus ANIs. This large difference in species/genus-level ANIs
provided evidence that these marker-gene-containing sequences were sufficient for delivering
species-level taxonomic resolution.

**Changes:**

We have added the following text to Section 2.2 and Fig. S5 to Supplementary Note 1:

Furthermore, for fast-growing species, genome copies might be significantly overestimated or underestimated if marker genes are centered around replication origins or termini (Fig. 2c). We selected a single representative RPG from each cluster to serve as a marker gene. This led to a total of eight marker genes for both bacteria and archaea.

We then extracted sequences of at most 10,000 bp that covered the selected marker genes from RefSeq assemblies. After deduplication, we computed their pairwise average nucleotide identities (ANIs) using skani (Shaw et al., 2023). The median ANI ranged from 99.0% (bacterial s16) to 99.8% (bacterial l2) between strains and from 93.8% (archaeal l2) to 96.9% (archaeal l10e) between species. Given that RPGs are in general more conserved compared to other nonessential genes, it was expected that some between-species ANIs exceeded the conventional ANI cutoff for species identification, i.e., 95%. However, for most marker genes, there was still a gap between the within-species and within-genus ANIs. This large difference in species/genus-level ANIs provided evidence that these marker-gene-containing sequences were sufficient for delivering species-level taxonomic resolution (Supplementary Note 1).

Of the 310,881 RefSeq assemblies, 284,039 (91.4%) possessed precisely eight unique marker genes.

**Comment 7**

I am very interested to see how the RPGs assessment goes in GTDB instead of RefSeq, considering that RefSeq has issues with confused taxonomy annotation. Since GTDB relies solely on genome similarity, it could potentially retain more markers, possibly enhancing abundance estimation with a larger pool of marker genes.

**Reply:**

Using GTDB for assessment is indeed a great idea as it already has a list of species-level
representative assemblies, selected not only according to completeness and contamination
scores but also to other criteria such as being assembled from type material (Parks et al.,

2022). We therefore reassessed the 91 RPGs using all complete genome representative
 assemblies ($n = 7,523$) of the latest GTDB (R220).

Figure R2 for reviewer 1: Selection of marker genes (GTDB). Mean copies, universality and overall performance of RPGs. Sizes of white dots represent the values of universality, with universality of one being omitted. Overall universality, deviance and $F_{0.5}$ -scores of RPGs are shown below the heatmaps of mean copies. Shapes of triangles indicate evaluation sets, with top-right triangles representing the full set of assemblies, and bottom-right triangles the tiny genome subset. RPGs satisfying all three criteria (darkest colors in all three rows) for either bacteria or archaea are marked in bold. The bar chart on the right side shows the number of assemblies (species) present in each phylum. Phyla with only a single assembly are grouped under “others”.

As the reviewer anticipated, with GTDB, seven more bacterial RPGs could pass the uni-
 versality and deviance cutoff, including s8, s10, l13, l17, l19, l22, and l24 (Fig. R2). There
 were no additionally selected RPGs for archaea. In fact, archaeal l18e was no longer se-
 lected as it was missed in one of the 13 tiny archaeal genomes (*ca. Micrarchaeota archaeon*,
 GCA_021654395.1, excluded by RefSeq due to (1) derived from metagenome and (2) genus
 undefined) for some reason. These additional RPGs were unfortunately found to be located
 closely to the previously selected RPGs, and after clustering based on their genomic distances,
 we still obtained eight clusters. As mentioned in Section 3, collocation is a known issue for
 RPGs, although they have the advantages of being essential and conserved. In the future,
 we might consider including genes other than RPGs, to expand the candidate list.

Figure R3 for reviewer 1: RPG assessments with NCBI and GTDB.

Upon closer inspection of the assessments we found that, although with GTDB more RPGs
 could be retained, the overall universality and deviance tended to be more pessimistic (Fig.
 R3). This could be caused by the fact that NCBI considers many unclassified species as
 individual species, even though they have a high ANI with known species (e.g. *Escherichia*
 *sp.*, taxonomic ID 1884818).

**Comment 8**

Line 231: the explanation of Bracken and mOTUs' performance is confusing. Please provide a clearer explanation and define what is meant by the recovery rate.

**Reply:**

The recovery rate shows the number of genome copies that could be detected by the
 corresponding profilers and is synonymous with the x/y ratio that we defined previously. We
 used these two terms interchangeably, with the exception that the recovery rate might refer
 to species-level genome copies, while the x/y ratio exclusively to total genome copies.

**Changes:**

We have added the definition of the recovery rate to Section 2.3.1:

However, as reflected by their inferior x/y ratios ~~and~~ (ratios between the estimated and expected total genome copies, i.e. recovery rate of genome

copies) and Pearson's correlations, the estimates of Kraken and Bracken tended to fluctuate around the expected values.

We have revised the text about the performance of Bracken and mOTUs:

Bracken's inconsistency likely stemmed from the assumption that all reads were of equal length, which introduced variance in genome size correction. For mOTUs, the discrepancy was probably due to varied recovery rates of different species, which skewed the abundance estimates.

To see the performance of Melon in more complex settings, we collected 64 mouse gut profiles from the second Critical Assessment of Metagenome Interpretation (CAMI) challenge (Meyer et al., 2022).

**Comment 9**

Line 237: what was the rationale behind choosing 500,000 reads for the simulation??

**Reply:**

The simulator we used (metagenome mode of NanoSim v3.1.0) does not support simulation
at a given sample size but requires users to specify a targeted number of reads. Since CAMI2
samples have a fixed number of bases (5 Gb), we chose 500,000 reads to ensure that the
maximum size of samples did not deviate too far from 5 Gb. The resulting simulated samples
had sizes ranging from 2.3 Gb to 5.3 Gb (depending on the models used for simulation).

**Comment 10**

Line 254: in which species richness? or are you using the Wilcoxon signed-rank test?

**Reply:**

Yes, we used the two-sided Wilcoxon signed-rank test to compare the dissimilarity metrics
(between ground truth and estimated relative abundances of species) returned by different
tools. Here we didn't compare the richness as it can be reflected by the precision-recall
plots: Kraken2 tended to have an inflated richness estimation since it had many false positive
species, while Melon might sometimes underestimate the richness due to its relatively low
recall.

**Comment 11**

Line 300: It's not just Kraken that faces the problem mentioned; it's a common issue for every metagenomic profiler. The discussion could instead focus on the type of relative abundance generated by DNA-to-DNA methods.

168

**Changes:**

We have revised the discussion in Section 2.3.3 to highlight the limitation of DNA-to-DNA profilers (including Kraken):

Lastly, we emphasize again that while ~~Kraken demonstrated DNA-to-DNA profilers (e.g., Kraken here) may demonstrate~~ good performance in estimating both genome copies and relative abundances, ~~its~~ ~~their~~ accuracy hinges on the comprehensiveness of the genome size database. In our simulation ~~experiment~~ ~~experiments~~, genome size information was available for all species, making it unsurprising that Kraken achieved commendable results. However, as many species in real-world metagenomes still lack complete genome representatives, deriving a precise genome size database for correction can be very challenging. Melon, in contrast, does not face this limitation.

**Comment 12**

Line 306: it is good to see the application of Melon, but adding some validation, such as using shotgun sequencing data and a well-established pipeline for ARG calculation as a comparison, would greatly enhance its credibility.

**Reply:**

We agree with the reviewer that the application of Melon lacks justification. We downloaded
and reanalysed the 109 Singaporean human faecal samples (paired Nanopore and Illumina
data) from PRJEB49168 (Singapore Platinum Metagenomes Project) (Gounot et al., 2022)
using both the strategy we presented in this manuscript and ARGs-OAP v3.2.4 (Yin et al.,
2022). ARGs-OAP is a short-read-based ARG quantification pipeline developed by our group.
Since ARGs-OAP does not provide species-level taxonomic information, we compared the total
ARG abundance (expressed as ARG copies per genome copy) of the samples. As shown in
Fig. S7, the estimated ARG abundances were highly correlated (Spearman's correlation
$\rho = 0.941$, permutation test, two-sided, $p < 0.001$).

Despite being highly correlated, the ARG abundances estimated by long reads were system-
atically higher due to: (1) the database we used in this manuscript was the *full* version of
the SARG database, which includes additional genes such as transcriptional regulators (e.g.,
activators and repressors) compared to ARGs-OAP's *short* version; (2) ARGs-OAP considers
two- or three-component systems (e.g., *arcA-arcB-tolC*, the primary efflux pump of *E. coli*)
as single units. These genes are weighted by a factor of either 1/2 or 1/3, leading to a
significant reduction in ARG abundance estimation; and (3) the cutoffs we employed for long
reads (identity 75% and subject cover 75%) may not be directly comparable to those used
by ARGs-OAP (identity 80% and query cover 85%).

We note that developing a long-read-based ARG quantification pipeline was not our primary
goal in this project. Instead, we aimed to demonstrate Melon’s ability to provide reliable
species-level genome copy estimates, which could be used as normalizing constants for ARGs.
Further evaluation and parameter tuning may be required to achieve more precise ARG
abundance estimation.

Figure S7: ARG abundances estimated using short and long reads. Colors and shapes indicate subjects’ age and gender, respectively.

Changes

We have added Fig. S7 to Supplementary Note S3 and the following text to Section 2.4.1 to demonstrate the validity of the method:

In a pilot test using mock sample S3 (D6300), we observed good congruence between estimated and expected species-level ARG abundances (Spearman’s correlation $\rho = 0.915$, permutation test, two-sided, $p < 0.001$, Supplementary Figure 3), despite some ARGs being underestimated due to identical cutoffs employed for both ONT reads and reference genomes (sequencing errors in ONT reads caused some boundary cases to be missed). Using 109 human faecal samples (sequenced by both Nanopore and Illumina) as an additional validation dataset (Gounot et al., 2022), we observed a high correlation between the total ARG abundances estimated by this approach and ARGs-OAP (Yin et al., 2022) (Spearman’s correlation $\rho = 0.941$, permutation test, two-sided, $p < 0.001$, Supplementary Note 3).

We then applied the method to two ONT samples: influent and effluent of a wastewater treatment plant located in Hong Kong (Shatin, 22.407° N, 114.214° E, Supplementary Tables 1 and 3).

**Comment 13**

Additionally, given that the primary difference between Melon and mOTUs is the selection of marker genes and the alignment of flanking sequences used in abundance estimation, I see no reason why this method couldn't also be applied to short sequencing reads. Perhaps this could be evaluated in the manuscript or considered for future work.

**Reply:**

Thank you for this suggestion. The marker gene database we constructed can indeed be
applied to short sequencing reads, and the *total* genome copies estimated should still be
accurate given appropriate alignment settings and tools. However, due to length limitations,
short reads may not be able to cover sufficient flanking regions and therefore cannot provide
detailed *species-level* genome copies. In fact, we plan to replace the single-copy marker gene
database of `ARGs_OAP` with the newly constructed protein database, but this requires more
detailed evaluation. We leave this for future work.

References

- Gounot, J.-S., M. Chia, D. Bertrand, W.-Y. Saw, A. Ravikrishnan, A. Low, Y. Ding, A. H. Q.
Ng, L. W. L. Tan, Y.-Y. Teo, et al. (2022). “Genome-centric analysis of short and long
read metagenomes reveals uncharacterized microbiome diversity in Southeast Asians”. In:
*Nature Communications* 13.1, p. 6044.
- Meyer, F., A. Fritz, Z.-L. Deng, D. Koslicki, T. R. Lesker, A. Gurevich, G. Robertson,
215 M. Alser, D. Antipov, F. Beghini, et al. (2022). “Critical assessment of metagenome
interpretation: the second round of challenges”. In: *Nature methods* 19.4, pp. 429–440.
- Parks, D. H., M. Chuvochina, C. Rinke, A. J. Mussig, P.-A. Chaumeil, and P. Hugenholtz
(2022). “GTDB: an ongoing census of bacterial and archaeal diversity through a phyloge-
netically consistent, rank normalized and complete genome-based taxonomy”. In: *Nucleic
acids research* 50.D1, pp. D785–D794.
- Portik, D. M., C. T. Brown, and N. T. Pierce-Ward (2022). “Evaluation of taxonomic
classification and profiling methods for long-read shotgun metagenomic sequencing datasets”.
In: *BMC bioinformatics* 23.1, p. 541.
- Ruscheweyh, H.-J., A. Milanese, L. Paoli, N. Karcher, Q. Clayssen, M. I. Keller, J. Wirbel,
P. Bork, D. R. Mende, G. Zeller, et al. (2022). “Cultivation-independent genomes greatly
expand taxonomic-profiling capabilities of mOTUs across various environments”. In: *Mi-
crobiome* 10.1, pp. 1–12.
- Shaw, J. and Y. W. Yu (2023). “Fast and robust metagenomic sequence comparison through
sparse chaining with skani”. In: *Nature Methods* 20.11, pp. 1661–1665.
- Sun, Z., S. Huang, M. Zhang, Q. Zhu, N. Haiminen, A. P. Carrieri, Y. Vázquez-Baeza, L.
Parida, H.-C. Kim, R. Knight, et al. (2021). “Challenges in benchmarking metagenomic
profilers”. In: *Nature methods* 18.6, pp. 618–626.
- Ye, S., K. Siddle, D. Park, and P. Sabeti (2019). “Benchmarking Metagenomics Tools for
Taxonomic Classification.” In: *Cell* 178.4, pp. 779–794.
- Yin, X., X. Zheng, L. Li, A.-N. Zhang, X.-T. Jiang, and T. Zhang (2022). “ARGs-OAP v3.
0: Antibiotic-resistance gene database curation and analysis pipeline optimization”. In:
*Engineering*.

**Reviewer 2**

The authors presented a new tool Melon for long-read based taxonomic profiling. The
authors justified the usage of marker-gene based method and tested Melon in both mock and
real data. My main concern is that while the authors described Melon as a profiler tailed
specifically for error-prone long reads, the design of the methodology and the benchmarking
experiments did not incorporate the properties of TGS or the tools that are specifically
designed for TGS. Thus, it is not clear whether the main contribution is the set-up of the
reference maker database or the pipeline.

**Author response:**

Thank you for the comments. We have added a new benchmark experiment to demonstrate
Melon's advantages over existing taxonomic profilers, including the long-read-specific ones
MetaMaps and MEGAN-LR. Additionally, we have revised Fig. 1 and rewritten Sections 2 and
5 to enhance the overall readability. Please see below our point-by-point responses.

**Major**

**Comment 1**

As there are previous research about single copy marker genes, have the authors compared their markers with those single copy genes? Seems using more single-copy marker genes is expected to achieve a higher recall while simplifying the abundance computational.

**Reply:**

We agree with the reviewer that including more marker genes will lead to a higher recall
and may work better for detecting rare species/pathogens. With the current marker gene
set (8 for bacteria, 8 for archaea), Melon is able to detect species with a coverage down to
0.125. Including more marker genes will indeed push this limit lower but also increase the
computational time linearly. For instance, to achieve a detection limit of 0.01, we would
need at least 100 marker genes, and the computation time would be scaled by a factor of
12.5 (approximately). This may lead to a computational burden.

We compared our marker genes with those from mOTUs (10), GTDB-Tk (120 for bacteria, 53
for archaea), and MicrobeCensus (30). GTDB-Tk and MicrobeCensus also include a large
proportion of RPGs (COG category J - Translation, ribosomal structure, and biogenesis),
thus having a high similarity with Melon. mOTUs does not incorporate any RPGs since its
taxonomic classification relies solely on the taxonomic labels of marker genes. Therefore, its
marker genes need to be as discriminative as possible and, consequently, these genes may not
be as essential as RPGs.

We also showed in Fig. 2c that, ideally, marker genes should have a balanced distribution
along the genome, and using more marker genes may not always lead to a more accurate

genome copy estimation. If most marker genes are located near the replication origin, the
 estimated genome copies could be overestimated for fast-growing species.

Figure 2c: Selection of marker genes. Effects of marker genes' locations on genome copy estimations. Black dots represent sequencing coverage at different genomic locations, smoothed using 5,000 bp non-overlapping windows. Colored dots display the genomic locations of marker genes in four sets: Melon (8 genes), GTDB (120 genes), mOTUs (10 genes), and MicrobeCensus (30 genes). Dashed black lines denote expected genome copies, whereas solid colored lines stand for estimated genome copies produced by averaging the coverage of marker genes in different sets, smoothed again using 5,000 bp windows. Melon's genome copy estimates are the closest to the expected genome copies of both *Bacillus amyloliquefaciens* (*B. amyloliquefaciens*, Gram-positive) and *Pseudomonas alloputida* (*P. alloputida*, Gram-negative), owing to its relatively balanced marker gene distributions. \log_2 -transformed peak-to-trough ratios (PTRs) are displayed in text. *OriC* means the origin of replication.

**Comment 2**

As long reads can contain insertion/deletion sequencing errors, I am wondering whether the authors did any error correction, which can affect the gene prediction and translation. Or is DIAMOND under some setting is very robust against sequencing errors? Did the authors consider using optimized tools for long read alignment?

**Reply:**

Yes, as long reads can have a large proportion of insertion/deletion errors the genes predicted
 (e.g., via Prodigal) from these reads may not be trustworthy. We didn't correct or assemble
 the reads but employed DIAMOND's *long-read mode blastx* to do the translated search directly.
 This long-read mode was introduced to DIAMOND in version 0.9.23 and, according to the
 developers, is capable of conducting *frame-shift aware* DNA-to-protein alignment (Arumugam
 et al., 2019). This mode is also used by MEGAN-LR to handle error-prone long reads.

We admit that we didn't look into the detailed parameter settings of this mode, as we
 empirically found the default parameters worked well for all the mock samples, which had a
 wide range of quality scores. As is shown in Fig. 3a, the total genome copies were obtained
 directly from DIAMOND's frame-shift aware alignment and were all close to the expected values.
 It is worth noting that we used DIAMOND as a filter for marker-gene-containing reads. The
 final species-level taxonomy was not given by DIAMOND's translated search (protein database)
 but by `minimap2`'s base-level alignment (nucleotide database).

Figure 3a: Performance of Melon on mock samples. Mock experiment (mock samples S1–3 and G1–3). Colors indicate tools used for comparison, including Melon (v0.1.0, NCBI database, ver. 2023-07-31), Kraken (v2.1.3, Standard database, ver. 2023-06-05), Bracken (v2.8, same database as Kraken) and mOTUs (v3.1.0, default database, with modifications). Shapes of points denote different mock communities, D6300 and D6331. Tables provide summarized metrics, including: ratio (x/y ratio), corr. (Pearson's correlation), pr. (precision), re. (recall), and F₁ (F₁-score). Abbreviations used for tools' names are: M. (Melon), K. (Kraken), m. (mOTUs), and B. (Bracken). Error bars depict the ranges of possible values. All metrics, except for Pearson's correlation, are aggregated by averaging where necessary.

**Comment 3**

Until Section 5.1.5, it is not obvious which part of the methodology design is optimized for long reads. In the experiments, the reviewer also expects to see the performance of Melon w.r.t. some data properties of TGS. For example, how does Melon's performance change with the read length, error rate, and coverage? Many TGS data may not be able to achieve a very high coverage. Seems these key factors are not evaluated.

**Reply:**

The major difference between **MelOn** and other marker-based methods (**mOTUs**, **MetaPhlAn**) is
that **MelOn**'s taxonomic labelling is based on marker genes plus their flanking regions, rather
than the marker genes themselves. Although this extension is straightforward, it allows
**MelOn** to take advantage of long reads and give more detailed species-level taxonomic labels
with high accuracy. In Fig. S2, we see that the flanking regions were indeed important and
there was an obvious improvement in precision and recall when the sequence length increased
from 5,000 to 10,000 bp. This implies that using marker genes alone (which are typically
below 1,000 bp) is not adequate for long-read taxonomic classification.

Figure S2: Performance of Melon at different length cutoffs of flanking regions. Models used for simulation were trained using six mock samples (S1–3 and G1–3). Each combination of simulation models and numbers of species contains ten randomly generated profiles. Colors represent the mean values of these profiles.

In that sense, **MelOn** can also be seen as a DNA-to-DNA profiler with a specialized database.

First, the incorporation of flanking regions allows **MelOn** to go beyond DNA-to-marker
 profiling and fully utilize the long-range information of long reads. Second, the single-copy
 and universal nature of marker genes enables **MelOn** to directly yield taxonomic abundance
 rather than sequence abundance (which requires genome-size correction to get taxonomic
 abundance). Third, the relatively small size of the database permits **MelOn** to profile against
 the entire collection of GTDB, which currently includes around 600k assemblies (R220), using
 base-level alignment on a standard laptop. In comparison, profiling against the entire GTDB
 with **Kraken** may require terabytes of memory, which is not considered computationally
 feasible for most users.

The properties of TGS, including coverage, quality score q , and read length l , are indeed
 important for evaluating long-read profilers. However, unlike Illumina simulators, TGS
 simulation tools such as **NanoSim** cannot directly simulate a series of samples with various
 q and l combinations but require error profiles to generate simulations. This is the main
 reason we collected six mock samples with a wide range of read characteristics (mean q
 10.9-19.0, mean l 4.1k-10.4k, Supplementary Table S3). We trained six different models using
 these mocks to mimic their read characteristics and used them for simulation instead. In
 the complexity experiment, we changed the number of species (32-512) and fixed the total
 number of reads to achieve different coverages. We saw that **MelOn** did not work well when
 the number of species was high (and in turn, the coverage of species was low), but this is a
 common limitation for marker-based methods.

Table S3: Statistics of mock and wastewater samples

platform	mock	sample	read	Gb ^a	length				quality score	
					longest	shortest	mean	median	mean	median
ONT	D6300	S1	2,503,848	16.619	185,010	1,000	6,637	3,947	10.899	11.037
		S2	2,119,258	9.514	51,733	1,000	4,489	3,673	12.085	12.044
		S3	336,330	3.486	211,938	1,000	10,365	3,879	13.187	13.320
	D6331	G1	5,650,723	27.891	50,325	1,000	4,935	4,569	13.742	14.243
		G2	1,570,234	6.458	51,516	1,000	4,112	3,581	16.987	17.321
		G3	713,445	3.587	38,817	1,000	5,028	4,171	19.045	19.365
	-	influent ^b	1,091,772	7.816	374,946	1,000	7,158	6,295	18.320	18.677
		effluent ^b	1,118,502	5.158	91,897	1,000	4,611	3,443	18.211	18.539
	PacBio	MSA-1003	-	2,418,889	20.544	21,547	1,001	8,493	8,310	38.768
D6331		-	1,978,476	17.993	39,601	1,003	9,094	8,078	45.521	39.623

^a Gigabase pair (10^9 base pairs).

^b Two spike-ins species *Allobacillus halotolerans* and *Imtechella halotolerans* were removed before calculation.

Nevertheless, to demonstrate the robustness of **MelOn** with respect to q and l , we cut mock
 G1 into 25 bins. As shown in Fig. R4, **MelOn** performed equally well for all q and l , except
 for $q > 20$, where the total number of bases ranged from 1.7 Mb to 17.4 Mb. Such low

coverage hindered EM from correcting false positive reads, leading to relatively low precision.
 Recall was also miserably low due to insufficient coverage. Both x/y ratio and Bray-Curtis
 were more varied, indicating again the poor performance of Melon at low coverage.

Figure R4 for reviewer 2: Performance of Melon with different q and l .

**Comment 4**

How important is the EM step? Seems its contribution is not evaluated in the Results Section. It is only first mentioned in Discussion.

**Reply:**

EM helps to eliminate false positive species and is important for achieving high precision.
 Although the EM step may skew the relative abundance estimates if multiple highly similar
 species are present, we believe in reality this negative effect is small (see Comment 5).

**Comment 5**

The chosen benchmark tools are not designed for long reads. Kraken is not specifically designed for long reads. Also, as the authors pointed out, BWA in mOTU is also not optimized for long reads. Thus, the comparison is not very fair. Although the authors mentioned that MetaMaps, MEGAN-LR, and BugSeq are “sequence-based”, can they be customized with different reference database? If so, you can replace their reference data with your protein database? Additionally, you can also directly run Melon against these tools and compare their final outputs.

**Reply:**

We thank the reviewer for raising this concern. We have included a new benchmark experiment
to demonstrate the advantages of **Melon** over existing profilers, including the long-read-specific
ones **MetaMaps** and **MEGAN-LR**, as well as the recently published one **Centrifuger** (Song et al.,
2024). We did not compare **BugSeq** since it is a commercial web platform that requires quotas.
We excluded **mOTUs** because it does not support building databases from scratch.

In order to make a fair comparison, we exclusively used RefSeq complete genomes (collected on
May 10, 2024) and rebuilt the database for **Kraken2**, **Centrifuger**, **MetaMaps**, and **minimap2**
(the database of **minimap2** is shared by **minimap2+BH**, **minimap2+EM**, and **MEGAN-LR**).

We collected the reference genomes of the top 24 pathogens from NCBI (<https://www.ncbi.nlm.nih.gov/pathogens/organisms/>) and assembled one genome from a wild-type *E. coli*
isolated from a wastewater treatment plant. All these genomes were present in the RefSeq
database, except for the genome of the wild-type *E. coli*. We simulated two samples with the
25 genomes (even taxonomic abundance) using **NanoSim**: high-quality (HQ, profile G3) and
low-quality (LQ, profile S3), each with 500,000 reads. We specifically used the marker-gene-
containing reads to see whether various taxonomic classification strategies could correctly
identify their taxonomy. For evaluation, we focused on the percentage of misclassified reads.
The types of misclassification include (1) true positive (the misclassified species is within
the 25 species), (2) false positive (the misclassified species is not among the 25 species), and
(3) unclassified (the read is not utilized by the classifier or the classification does not have a
species-level taxonomy).

Regarding the taxonomic classification strategies, **minimap2** refers to the native base-level
alignment of Nanopore reads (preset **map-ont**). **minimap2+BH** retains a single best hit for
each read based on the alignment score (AS) and adopts its taxonomy. **minimap2+EM** is the
taxonomic assignment strategy of **Melon**, where taxonomic labels are reassigned with EM.
**MEGAN-LR** aggregates the alignments of **minimap2** using a lowest common ancestor (LCA)
algorithm called interval-union LCA (note that **MEGAN-LR** can run with either a protein or a
nucleotide database and by default uses **DIAMOND** with NCBI nr and **minimap2** with NCBI
nt). **MetaMaps** employs minimizer-based approximate mapping and EM for post-correction.
**Kraken2** and **Centrifuger** are both *k*-mer-based (or similar) alignment-free methods.

In Fig. S6, we see that **minimap2+BH** and **minimap2+EM** clearly outperformed all the other
methods. **MEGAN-LR** also employs **minimap2** but its LCA algorithm made many reads unclassi-
fied at species level. **Kraken2** showed the worst performance and its accuracy varied strongly
across species. Unlike **Kraken2**, **Centrifuger** performed reasonably well despite also being
alignment-free. **MetaMaps** could not correctly classify certain species, e.g., *Bacillus cereus* and
*Staphylococcus aureus*, resulting in much worse overall performance compared to **minimap2+EM**
(note that the implementation of EM may not be exactly identical). Interestingly, we observed
that a higher quality of reads did not always lead to better classification for all classifiers

except for **Centrifuger**. This suggests that these classifiers are less sensitive to quality score
 q but more sensitive to read length l . We also see that the additional EM step (**minimap2+EM**)
 greatly reduced the number of false-positive misclassifications but might accidentally skew
 the relative abundance estimate by introducing true-positive misclassifications due to the
 presence of highly similar species (e.g., *Neisseria meningitidis* being classified as *Neisseria*
 *gonorrhoeae*, ANI 94.94%). This experiment indicated the advantages of using base-level
 alignment plus post-error correction for accurate taxonomic labelling of long reads.

We didn't add **MetaMaps** and **MEGAN-LR** to the main text by rerunning the mock/simulation
 experiments due to resource limitations. We note that although **MetaMaps** uses approximate
 rather than base-level mapping, for the marker-gene-containing sequences that we simulated
 (approximately 1 Gb), it took over a day to finish on a lab-scale server (40-thread CPU, 500
 GB memory), possibly due to poor implementation. **MEGAN-LR** by default requires mapping
 against NCBI nr or nt, which is also computationally costly.

Figure S6: Comparison of taxonomic assignment strategies. **a.** Species-level misclassification. Color indicates the percentage of misclassified reads (Mis. %). Wild-type *E. coli* is labelled with an asterisk. **b.** Overall misclassification. Color represents types of misclassification. Characteristics of simulation profiles are shown in tables.

**Changes**

We have included this experiment to Supplementary Note S2 and added the following text to Section 2.3.2:

In precision-recall analysis, Melon achieved the highest precision with only a minor drop in recall. The high precision of Melon was likely attributable to the implementation of expectation-maximization (EM) as a post-correction module, which greatly reduced the number of false positive classifications (Supplementary Note 2). All other tools showed increased F_1 -scores, though to varied degrees.

**Comment 6**

A minor comment is related to the names of the two methods. After all, marker genes are still sequences. So using “sequence-based” to exclude marker-based tools can be confusing. Maybe “genome-based” is a better name.

**Reply:**

We thank the reviewer for this suggestion. We have changed all *sequence-based* to *DNA-to-*
*DNA/protein*, and most *marker-based* to *DNA-to-marker* to avoid potential confusion caused
by the names. This *DNA-to-DNA/protein/marker* notation was adopted from Sun et al.,
2021 and Ye et al., 2019.

**Changes:**

Here is one example of the changes in Section 1:

Depending on the type of the database employed, these classification methods can be broadly categorized into two groups: (1) ~~sequence-based~~ *DNA-to-DNA/protein* and (2) ~~marker-based~~ *DNA-to-marker* taxonomic profilers (Sun et al., 2021; Ye et al., 2019).

**Comment 7**

Method 5.1.1 It will be clearer to describe the change of the number of sequences w.r.t. the multiple “filtration” strategies. In particular, it is not clear what is the purpose of the second alignment with full KOfam database. Also, is the scaled cutoff applied to both the 91 and 11,298 pHMMS or just the latter? The described procedure is hard to reproduce.

**Reply:**

The main reason we performed two rounds of `hmmsearch` is that some protein sequences
may cover more than one domain (possibly due to mutations in the stop codons or simply

misassembly; for instance, protein WP_109570822.1 is fused by a ribosomal protein s2 and a
 translation elongation factor). These protein sequences cannot be detected using only the
 91 RPG profiles and are generally favored by alignment tools due to their longer lengths.
 We want to minimize the effects of these proteins but acknowledge that the number of these
 proteins is rather small ($n = 38,568$) and their effects may be negligible.

The scaled cutoff was applied to both the 91 and the 11,298 PHMMs. We justified the choice
 of 0.75 in Supplementary Table S2.

Table S2: Performance of PHMMs at different scales of threshold scores

	scale	precision	recall	F _{0.5} -score	F ₁ -score	RPGF ^a
bacteria	0.50	0.969	0.931	0.954	0.942	42
	0.75	0.983	0.961	0.977	0.969	45
	1.00	0.984	0.958	0.977	0.968	45
archaea	0.50	0.920	0.677	0.800	0.734	21
	0.75	0.936	0.703	0.809	0.749	23
	1.00	0.906	0.639	0.750	0.688	19

^a Number of RPGFs with PHMM’s F_{0.5}-scores greater than 0.99.

**Changes:**

We have added the number of sequences for each step to Section 5.1.1:

Raw protein sequences (nr and env_nr, $n = 606,124,298$) were retrieved from the NCBI FTP server (<https://ftp.ncbi.nlm.nih.gov/blast/db>) on July 31, 2023. Bacterial and archaeal sequences ($n = 473,274,163$) were extracted from the nr database using TaxonKit v0.14.3 (Shen et al., 2021) and the ‘blastdbcmd’ command from BLAST+ v2.14.0 (Camacho et al., 2009). Sequences shared by both bacteria and archaea ($n = 67,318$) were discarded to avoid cross-mapping. For sequences of env_nr ($n = 10,951,228$), taxonomic information was not available. We thus mapped these sequences to the full nr database using DIAMOND v2.1.8 (Buchfink et al., 2021) (‘blastp --top 0’), and subsequently removed every sequence whose best-scoring hits had a least common ancestor that was either unknown or non-prokaryotic ($n = 294,083$).

Remaining sequences ($n = 483,863,990$) were functionally annotated using the ‘hmmsearch’ command from HMMER v3.3.2 (Aramaki et al., 2020) and the KOfam PHMMs database (ver. 2023-04-01). To reduce computational time, the annotation was carried out in two steps: we first extracted sequences that putatively contained at least one RPG using a subset of the KOfam database (91 PHMMs), and then annotated the selected sequences ($n = 3,927,293$)

using the full KOfam database (11,298 PHMMs). The pre-computed adaptive score thresholds associated with PHMMs were scaled by a factor of 0.75 to prevent dropping too many RPGs. To rule out spurious hits, sequences to which more than one PHMM was aligned (non-unique hits, $n = 38,568$) or whose length was covered by less than or equal to 75% by all aligned PHMMs (incomplete hits, $n = 199,811$), were discarded. With this, ~~over three million~~ 3,688,914 sequences were classified as RPGs. To reduce redundancy, these sequences were further clustered using MMseqs2 v14.7e284 (Steinegger et al., 2017) ('easy-cluster -s 7.5 -c 0.98 --min-seq-id 0.98 --cov-mode 0 --cluster-reassign'), leading to a total number of 2,850,814 sequences.

**Comment 8**

Overall, I found the descriptions at several places are hard to follow partly because of the scattered description of the major steps in different places and partly because the description is like a laundry list lacking justification. Two examples are given below. The second paragraph of Section 2.1 is a little hard to follow. The authors said “four factors are considered”. But there lacks clear description of how to reduce 91 pHMMs to 8 pHMMs and how each factor is used specifically. Later, seems ‘8’ refers to only 8 marker gene not 8 gene clusters (pHMMs) in 5.1.3.

**Reply:**

We apologize for the lack of clarity. Our intention was to provide an introductory overview
of MeLon and its database (i.e., description of Fig. 1) in Section 2.1 before moving on to
marker-gene selection in Section 2.2. Therefore, it was inevitable to mention the “four factors”
in both Sections 2.1 and 2.2, even though the detailed procedure for reducing the 91 PHMMs
to 8 PHMM clusters using these factors and selecting marker genes from each cluster was
described only in Section 2.2. We agree that this might cause confusion when reading in the
current order. To address this, we have added a reminder to guide readers more clearly.

**Changes:**

We have added a reminder before the “four factors” in Section 2.1:

A subset of RPGs were screened as marker genes (eight each for both bacteria and archaea, see section “Quality assessment of PHMMs and RPGs” for more details) by assessing the quality of each according to four distinct factors: (1) $F_{0.5}$ -scores ~~of~~ (weighted harmonic mean of precision and recall) of their associated PHMMs, (2) their prevalence among species, (3) discrepancies between their average numbers of copies and the ideal count of one, and (4) their mean relative genomic distances to other candidate RPGs (Fig. 1c).

**Comment 9**

Another example. Up to Page 10, it is still not clear how the authors identify reads that cover marker genes. The authors mentioned “Melon first extracts reads that cover at least one marker gene using a protein database”. Fig. 1 said “Aligning sequences of assemblies to the marker gene database”. Neither of “extract” nor “align” is very specific. In 5.1.5, the authors mentioned the usage of “DIAMOND” for aligning RefSeq assemblies with the protein database. Better highlight the used method/tool for the major steps when those steps are introduced.

**Reply:**

We opted not to mention the tools in Section 2.1 since we wanted to keep the introductory
overview simple and non-technical. Detailed descriptions of the tools, including their versions
and commands, were given in Section 5.1 (Materials and methods). Nevertheless, we have
added the names of the main tools to Fig. 1 and hope that this will reduce the confusion.

**Changes:**

We have revised Fig. 1 by adding tools’ names for the major steps:

Fig. 1 Overview of Melon. a. Melon’s workflow. Given a sequenced metagenomic sample, Melon first extracts reads that cover at least one marker gene using a protein database (DIAMOND), and then profiles the taxonomy of these marker-containing reads using a nucleotide database (minimap2). The main output of Melon is a tab-delimited table listing the estimates of species’ genome copies and relative abundances. Grey dashed arrows and text indicate necessary sample preprocessing steps to obtain metagenomic long reads. **b.** The construction of the protein database is initiated by re-annotating NCBI protein sequences using hmmsearch and a set of RPG-related PHHMs. **c.** A subset of RPGs are selected as marker genes based on their universality, deviance, $F_{0.5}$ -scores, and mean relative genomic distances. **d.** The nucleotide database is built by extracting 10,000 bp genomic regions, which encompass marker genes and their adjacent flanking regions, from 310,881 RefSeq assemblies using DIAMOND and SeqKit. These assemblies represent 44,057 bacterial and 1,016 archaeal species. **a-d.** Solid colored dots stand for marker genes, while semi-transparent grey dots represent other genes. Circular elements denote genomes, while linear elements signify reads.

Figure 1: Overview of Melon. **a.** Melon's workflow. Given a sequenced metagenomic sample, Melon first extracts reads that cover at least one marker gene using a protein database (DIAMOND), and then profiles the taxonomy of these marker-containing reads using a nucleotide database (minimap2). The main output of Melon is a tab-delimited table listing the estimates of species' genome copies and relative abundances. Grey dashed arrows and text indicate necessary sample preprocessing steps to obtain metagenomic long reads. **b.** The construction of the protein database is initiated by re-annotating NCBI protein sequences using hmmsearch and a set of RPG-related PHHMs. **c.** A subset of RPGs are selected as marker genes based on their universality, deviance, $F_{0.5}$ -scores, and mean relative genomic distances. **d.** The nucleotide database is built by extracting 10,000 bp genomic regions, which encompass marker genes and their adjacent flanking regions, from 310,881 RefSeq assemblies using DIAMOND and SeqKit. These assemblies represent 44,057 bacterial and 1,016 archaeal species. **a-d.** Solid colored dots stand for marker genes, while semi-transparent grey dots represent other genes. Circular elements denote genomes, while linear elements signify reads.

**Comment 10**

There are a large number of cutoffs/parameters. Is there a better way to present them and highlight the justifications?

**Reply:**

Thank you for pointing this out. We agree that a large number of cutoffs and parameters
 were mentioned in this manuscript. Here is a brief summary of all the major cutoffs and
 parameters we considered (Table R1):

Table R1 for reviewer 2: Summary of parameters and cutoffs

Parameter/cutoff	Value	Description	D	T	J	E ^a
DIAMOND for extracting marker-containing sequences/reads or annotating ARGs						
--evaluate	10 ⁻¹⁵	max. expected value		×		×
--subject-cover	75	min. cover of subject sequences in percentage		×		×
--id	0	min. alignment identity in percentage (marker genes)	×	×		×
--id	75	min. alignment identity in percentage (ARGs)				×
--max-target-seqs	25	max. number of target sequences to report	×	×		×
--max-hsps	0	max. number of HSPs per target sequence to report				×
--range-culling	-	restrict hit culling to overlapping query ranges	×			×
--frameshift	15	frameshift penalty	×			×
--range-cover	25	percentage of query range to be covered for range culling				×
minimap2 for assigning taxonomic labels to marker-gene-containing reads						
-f	0	top fraction of most frequent minimizers to ignore				×
-p	0.9	min. secondary/primary score ratio to report secondary alignments		×		×
-N	2 ³¹ - 1	max. number of secondary alignments to report		×		×
mmseqs2 for clustering of protein/nucleotide databases						
-s	7.5	clustering sensitivity				×
--cluster-reassign	-	reassign cluster labels after cascaded clustering				×
--cov-mode	0	clustering cover mode (protein)	×			×
--cov-mode	1	clustering cover mode (nucleotide)				×
-c	0.98	min. fraction aligned (protein)				×
-c	0.9998	min. fraction aligned (nucleotide)				×
--min-seq-id	0.98	min. sequence identity (protein)				×
--min-seq-id	0.9998	min. sequence identity (nucleotide)				×
EM associated parameters						
ϵ	10 ⁻⁵	epsilon (precision)		×		×
t_{\max}	100	max. number of iterations		×		×
Database associated parameters						
length	10000	max. length of marker-gene-containing sequences			×	
Cutoffs for RPGs annotation						
cover	75	percentage of target sequences covered by PHMMs				×
scale	0.75	scale of Kofam's score thresholds			×	
Cutoffs for marker-gene selection						
universality	0.99	proportion of assemblies containing a gene				×
deviance	0.99	absolute difference between a gene's mean copy and one				×
F _{0.5} -score	0.99	F _{0.5} -score of a gene's PHMMs				×
distance	0.05	mean relative genomic distance between genes				×

^a D: Default. T: Tunable. J: Justified. E: Empirical.

We provided justifications for the most relevant parameters (e.g., max. length of marker-
gene-containing sequences, scale of Kofam’s score thresholds). Some parameters are given
by the corresponding tools and can be tuned by users (e.g., `--max-target-seqs`). Other
parameters are rather empirical (e.g., `--min-seq-id`). We tried to use the default parameters
as much as possible, but exceptions did occur. We note that it is common to have multiple
parameters in profiling tools, and they are relevant to achieving high accuracy. For in-
stance, in the “Taxonomic-Profiling-Minimap-Megan” pipeline provided by PacBio ([https://](https://github.com/PacificBiosciences/pb-metagenomics-tools)
github.com/PacificBiosciences/pb-metagenomics-tools), `minimap2` was used with 12
tuned parameters: `-k 19 -w 10 -I 10G -g 5000 -r 2000 -N 100 --l-j-min-ratio 0.5`
`-A 2 -B 5 -O 5,56 -E 4,1 -z 400,50`. Evaluating all parameter combinations of various
tools using mock or simulated samples would require a substantial amount of effort. We
leave this to further studies.

**Minor**

**Comment 11**

Does the last sentence mean that the authors applied the 8 pHMMs to all nr and env-nr proteins and kept the aligned proteins? Reading the detailed methods can help understand these descriptions. But reading the paper following the given order can be a little confusing.

**Reply:**

Yes, the protein database (before deduplication through clustering) contained all nr and
env_nr sequences that aligned with the PHMMs of the selected marker genes. We have
added a sentence to make this clear.

**Changes:**

A clarifying sentence has been added to Section 2.1:

A subset of RPGs were screened as marker genes (eight each for both bacteria and archaea, see section “Quality assessment of PHMMs and RPGs” for more details) by assessing the quality of each according to four distinct factors: (1) $F_{0.5}$ -scores ~~of~~ (weighted harmonic mean of precision and recall) of their associated PHMMs, (2) their prevalence among species, (3) discrepancies between their average numbers of copies and the ideal count of one, and (4) their mean relative genomic distances to other candidate RPGs (Fig. 1c). ~~After deduplication via~~ Protein sequences of nr and env_nr that aligned with the PHMMs of these marker genes were retained for further analysis. ~~After deduplication through~~ clustering, the final protein database ~~contained~~ comprised 468,432 ~~marker gene unique~~ sequences.

**Comment 12**

Melon is a computational tool. Thus, Fig. 1 a. is a little confusing as it contains sampling, DNA extraction, sequencing.

**Reply:**

We thank the reviewer for noting this. We agree that these words in Fig. 1a can cause
confusion. However, removing the upper-left two subfigures would require reformatting the
layout of the entire figure. To mitigate this, we have changed the arrows of these necessary
sample preprocessing steps from “solid” to “dashed” and the annotations from “black” to
“grey”. The caption has also been updated for clarity (see Comment 9).

**Comment 13**

2.3.2 please specify the tool you used to simulate TGS data at its first mention.

**Changes:**

We have added the simulation tool `NanoSim` to section 2.3.2:

For each profile, we simulated six samples (500,000 reads each) using the metagenome mode of NanoSim (Yang et al., 2023), with models trained specifically on the six mocks to emulate their error distributions and read characteristics (Supplementary Table 4).

References

- Aramaki, T., R. Blanc-Mathieu, H. Endo, K. Ohkubo, M. Kanehisa, S. Goto, and H. Ogata
(2020). “KofamKOALA: KEGG Ortholog assignment based on profile HMM and adaptive
score threshold”. In: *Bioinformatics* 36.7, pp. 2251–2252.
- Arumugam, K., C. Bağcı, I. Bessarab, S. Beier, B. Buchfink, A. Górska, G. Qiu, D. H. Huson,
and R. B. Williams (2019). “Annotated bacterial chromosomes from frame-shift-corrected
long-read metagenomic data”. In: *Microbiome* 7, pp. 1–13.
- Buchfink, B., K. Reuter, and H.-G. Drost (2021). “Sensitive protein alignments at tree-of-life
scale using DIAMOND”. In: *Nature methods* 18.4, pp. 366–368.
- Camacho, C., G. Coulouris, V. Avagyan, N. Ma, J. Papadopoulos, K. Bealer, and T. L.
Madden (2009). “BLAST+: architecture and applications”. In: *BMC bioinformatics* 10,
pp. 1–9.
- Shen, W. and H. Ren (2021). “TaxonKit: A practical and efficient NCBI taxonomy toolkit”.
In: *Journal of Genetics and Genomics* 48.9, pp. 844–850.
- Song, L. and B. Langmead (2024). “Centrifuger: lossless compression of microbial genomes
for efficient and accurate metagenomic sequence classification”. In: *Genome Biology* 25.1,
p. 106.
- Steinegger, M. and J. Söding (2017). “MMseqs2 enables sensitive protein sequence searching
for the analysis of massive data sets”. In: *Nature biotechnology* 35.11, pp. 1026–1028.
- Sun, Z., S. Huang, M. Zhang, Q. Zhu, N. Haiminen, A. P. Carrieri, Y. Vázquez-Baeza, L.
Parida, H.-C. Kim, R. Knight, et al. (2021). “Challenges in benchmarking metagenomic
profilers”. In: *Nature methods* 18.6, pp. 618–626.
- Yang, C., T. Lo, K. M. Nip, S. Hafezqorani, R. L. Warren, and I. Birol (2023). “Characteri-
zation and simulation of metagenomic nanopore sequencing data with Meta-NanoSim”. In:
*GigaScience* 12, giad013.
- Ye, S., K. Siddle, D. Park, and P. Sabeti (2019). “Benchmarking Metagenomics Tools for
Taxonomic Classification.” In: *Cell* 178.4, pp. 779–794.

2nd round

Reviewer 1

I am pleased to re-review the revised manuscript and acknowledge significant improvements made in response to the reviewers' earlier comments. I believe all the issues I previously raised have been effectively addressed, resulting in a much-enhanced version of the paper. I appreciate the authors' efforts in making these revisions.

Reviewer 2

The authors have addressed my comments well. The additional data/results can help readers better understand the design and appreciate the tool. No further questions.